# PERFECT ALIGNMENT MAY BE POISONOUS TO GRAPH CONTRASTIVE LEARNING

## ABSTRACT

Graph Contrastive Learning (GCL) aims to learn node representations by aligning positive pairs and separating negative ones. However, limited research has been conducted on the inner law behind specific augmentations used in graph-based learning. What kind of augmentation will help downstream performance, how does contrastive learning actually influence downstream tasks, and why the magnitude of augmentation matters? This paper seeks to address these questions by establishing a connection between augmentation and downstream performance, as well as by investigating the generalization of contrastive learning. Our findings reveal that GCL contributes to downstream tasks mainly by separating different classes rather than gathering nodes of the same class. So perfect alignment and augmentation overlap which draw all intra-class samples the same can not explain the success of contrastive learning. Then in order to comprehend how augmentation aids the contrastive learning process, we conduct further investigations into its generalization, finding that perfect alignment that draw positive pair the same could help contrastive loss but is poisonous to generalization, on the contrary, imperfect alignment enhances the model's generalization ability. We analyse the result by information theory and graph spectrum theory respectively, and propose two simple but effective methods to verify the theories. The two methods could be easily applied to various GCL algorithms and extensive experiments are conducted to prove its effectiveness.

## 1 INTRODUCTION

Graph Neural Networks (GNNs) have been successfully applied in various fields such as recommendation systems (He et al., 2020b), drug discovery (Liu et al., 2018), and traffic analysis (Wu et al., 2019), etc. However, most GNNs require labeled data for training, which may not always be available or easily accessible. To address this issue, Graph Contrastive Learning (GCL), which does not rely on labels, has gained popularity as a new approach to graph representation learning (Veličković et al., 2018; You et al., 2020).

GCL often generates new graph views through data augmentation (Chen et al., 2020; Zhu et al., 2020). GCL uses nodes augmented from the same node as positive samples and other nodes as negative samples, then maximize similarity between positive samples and minimize similarity between negative ones (Wang & Isola, 2020; Hassani & Khasahmadi, 2020). Data augmentation can be categorized into three types (Zhao et al., 2022): random augmentation (Veličković et al., 2018; Zhu et al., 2020), rule-based augmentation (Zhu et al., 2021; Wei et al., 2023; Liu et al., 2022), and learning-based augmentation (Suresh et al., 2021; Jiang et al., 2019). For instance, Zhu et al. (2020) randomly masks node attributes and edges in graph data to obtain augmented graphs; Zhu et al. (2021) uses node degree to measure its importance and mask those unimportant with higher probability; And Suresh et al. (2021) uses a model to learn the best augmentation and remove irrelevant information as much as possible. However, most data augmentation algorithms are designed heuristically, and there is a lack of theoretical analysis on how these methods will influence the downstream performance.

Some researchers have explored the generalization ability of contrastive learning (Arora et al., 2019; Wang & Isola, 2020; Huang et al., 2021). They propose contrastive learning works by gathering positive pairs and separating negative samples uniformly. Wang et al. (2022b) argues that perfect

alignment and uniformity alone cannot guarantee optimal performance. They propose that through stronger augmentation, there will be support overlap between different intra-class samples, which is called augmentation overlap (Saunshi et al., 2022; Huang et al., 2021). Thus, the alignment of positive samples will also cluster all the intra-class samples together, and lead to class-separated representations due to the limited inter-class overlap. However, Saunshi et al. (2022) points out that augmentation overlap may be relatively rare despite the excellent performance of contrastive learning methods. Hence, chances are that the success of contrastive learning cannot be solely attributed to alignment and augmentation overlap. It is of vital importance to evaluate how augmentation works in the contrastive learning process, why the magnitude of augmentation matters so much and how to perform better augmentation? As data augmentation on graphs could be more customized and the magnitude of augmentation can be clearly represented by the number of modified edges/nodes (You et al., 2020), we mainly study the augmentation on graphs. But it works the same in other fields.

In this paper, we provide a new understanding of Graph Contrastive Learning and use a theoretical approach to analyze the impact of augmentation on model generalization. We find that with a better augmentation, the model is performing better mainly because of inter-class separating rather than intra-class gathering brought by augmentation overlap. So perfect augmentation overlap and alignment are not the key factor for contrastive learning. To further analyze the phenomena, we formulate a relationship between downstream performance, contrastive learning loss, and augmentation, reveal the reason why stronger augmentation helps class separating, and find stronger augmentation could benefit the generalization by weaken the positive pair alignment. Therefore, perfect alignment is not the key to success, and may be poisonous to contrastive learning.

Then aiming to achieve a better balance between generalization and contrastive loss, we further analyze the contrastive process through information theory and graph spectrum theory. From the information theory perspective, we find augmentation should be stronger while reducing the information loss, which is actually adopted explicitly or implicitly by designed algorithms (Zhu et al., 2021; 2020; Suresh et al., 2021). From the graph spectrum theory perspective, we analyze how the graph spectrum will affect the contrastive loss and generalization (Liu et al., 2022), finding that non-smooth spectral distribution will have a negative impact on generalization. Then we propose two methods based on the theories to verify our findings.

Our main contributions are as follows. (1) We reveal that when stronger augmentation is applied, contrastive learning benefits from inter-class separating more than intra-class gathering, and imperfect alignment could be more beneficial as it enlarges the inter-class distance. (2) We establish the relationship between downstream performance, contrastive learning loss, and data augmentation. Further explains why stronger augmentation helps, then we analyze the result from information theory and graph spectrum theory to guide algorithm design. (3) Based on the proposed theoretical results, we provide specific algorithms that verify the correctness of the theory. We also show that these algorithms can be extended to various contrastive learning methods to enhance their performance. (4) Extensive experiments are conducted on different contrastive learning algorithms and datasets using our proposed methods to demonstrate its effectiveness.

## 2 AUGMENTATION AND GENERALIZATION

### 2.1 PRELIMINARIES

A graph can be represented as $\mathcal{G} = (\mathcal{V}, \mathcal{E})$, where $\mathcal{V}$ is the set of $N$ nodes and $\mathcal{E} \subseteq \mathcal{V} \times \mathcal{V}$ is the edge set. The feature matrix and the adjacency matrix are denoted as $\boldsymbol{X} \in \mathbb{R}^{N \times F}$ and $\boldsymbol{A} \in \{0, 1\}^{N \times N}$, where $F$ is the dimension of input feature, $\boldsymbol{x}_i \in \mathbb{R}^F$ is the feature of node $v_i$ and $\boldsymbol{A}_{i,j} = 1$ iff $(v_i, v_j) \in \mathcal{E}$. The node degree matrix $\boldsymbol{D} = \mathrm{diag}(d_1, d_2, ..., d_N)$, where $d_i$ is the degree of node $v_i$.

In contrastive learning, data augmentation is used to create new graphs $\mathcal{G}^1, \mathcal{G}^2 \in \mathbb{G}^{\mathrm{aug}}$, and the corresponding nodes, edges, and adjacency matrices are denoted as $\mathcal{V}^1, \mathcal{E}^1, \boldsymbol{A}^1, \mathcal{V}^2, \mathcal{E}^2, \boldsymbol{A}^2$. In the following of the paper, $v$ is used to represent all nodes including the original nodes and the augmented nodes; $v_i^+$ is used to represent the augmented nodes including both $v_i^1$ and $v_i^2$; $v_i^0$ represents the original nodes only.

Nodes augmented from the same one, such as $(v_i^1, v_i^2)$, are considered as positive pairs, while others are considered as negative pairs. It is worth noting that a negative pair could come from the same

graph, for node $v_i^1$, its negative pair could be $v_i^- \in \{v_j^+ | j \neq i\}$. Graph Contrastive Learning (GCL) is a method to learn an encoder that draws the embeddings of positive pairs similar and makes negative ones dissimilar (Chen et al., 2020; Wang & Isola, 2020). The encoder calculates the embedding of node $v_i$ by $f(\boldsymbol{X}, \boldsymbol{A})[i]$, which can be summarized as $f(v_i)$ for better comprehension, we assume that $||f(v_i)|| = 1$. The commonly used InfoNCE loss (Zhu et al., 2020) can be formulated as follows:

$$\mathcal{L}_{\text{NCE}} = \mathbb{E}_{p(v_i^1, v_i^2)} \mathbb{E}_{\{p(v_i^-)\}} \left[ -\log \frac{\exp(f(v_i^1)^T f(v_i^2))}{\sum_{i=1}^M \exp(f(v_i^1)^T f(v_i^-))} \right]. \tag{1}$$

We use two augmented views to perform GCL (Zhu et al., 2020; Chen et al., 2020). However, $v_i^1$ could be replaced by $v_i^0$ (Liu et al., 2022; He et al., 2020a), as $v_i^0$ is a special case of $v_i^1$ where the augmentation happen to change nothing about the original view.

## 2.2 HOW AUGMENTATION INFECT DOWNSTREAM PERFORMANCE

Previous work (Wang & Isola, 2020) proposes that effective contrastive learning should satisfy alignment and uniformity, meaning that positive samples should have similar embeddings, *i.e.,* $f(v_i^1) \approx f(v_i^2)$, and features should be uniformly distributed in the unit hypersphere. However, Wang et al. (2022b) pointed out that when $\{f(v_i^0)\}_{i=1}^N$ are uniformly distributed and $f(v_i^0) = f(v_i^+)$, there is a chance that the model may converge to a trivial solution that only projects very similar features to the same embedding, and projects other features randomly, then it will perform random classification in downstream tasks although it achieves perfect alignment and uniformity.

Wang et al. (2022b) argues that perfect alignment and intra-class augmentation overlap would be the best solution. The augmentation overlap means support overlap between different intra-class samples, and stronger augmentation would more likely bring more augmentation overlap. If two intra-class samples have augmentation overlap, then the best solution is projecting the two samples and their augmentation to the same embedding, which is called perfect alignment. For example, if two different nodes $v_i^0$, $v_j^0$ get the same augmentation $v^+$, then the best solution to contrative learning is $f(v_i^0) = f(v^+) = f(v_j^0)$. So perfect alignment and augmentation overlap could project all intra-class nodes to the same embedding, and project inter-class nodes differently because of limited inter-class overlap.

However, a stronger augmentation would connect more intra-class nodes by overlap, but will inevitably make achieving perfect alignment more challenging. Conversely a weaker augmentation would help alignment but augmentation overlap would be rare. Therefore, it is difficult to achieve both optimal augmentation and perfect alignment. And Saunshi et al. (2022) proposes that augmentation overlap is actually rare in practice, even with quite strong augmentation. So the better performance may not be brought by perfect alignment and overlap, in order to further study how exactly augmentation helps contrastive learning, we evaluate how the downstream performance changes while the augmentation being stronger.

To begin with, we give an assumption on the label consistency between positive samples, which means the class label does not change after augmentation.

**Assumption 2.1** (View Invariance). *For node $v_i^0$, the corresponding augmentation nodes $v_i^+$ get consistent labels: $p(y|v_i^0) = p(y|v_i^+)$.*

This assumption is widely adopted (Arora et al., 2019; Wang et al., 2022b; Saunshi et al., 2022) and reasonable. If the augmentation still keeps the basic structure and most of feature information is kept, the class label would not likely to change. Else if the augmentation destroys basic label information, the model tends to learn a trivial solution, so it is meaningless and we do not discuss the situation. Actually the graph data keeps great label consistency as discussed in Appendix C.9.

To further understand how is data augmentation is working in contrastive learning, we use graph edit distance (GED) to denote the magnitude of augmentation, Trivedi et al. (2022) proposes that all allowable augmentations can be expressed using GED which is defined as minimum cost of graph edition (node insertion, node deletion, edge deletion, feature transformation) transforming graph $\mathcal{G}^0$ to $\mathcal{G}^+$. So a stronger augmentation could be defined as augmentation with a larger GED.

**Assumption 2.2** (Positive Pair difference and Augmentation). *While Assumption 2.1 holds i.e., $p(y|v_i^0) = p(y|v_i^+)$, as the augmentation getting stronger, positive pair difference $\delta_{aug}^2 =$*

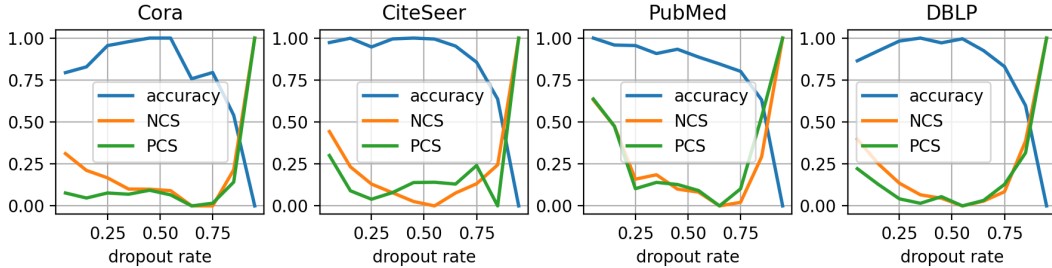

Figure 1: PCS means positive center similarity ($f(v_y)^T \mu_y$), NCS means negative center similarity ($f(v_y)^T \mu_{y^-}$) and accuracy is the downstream performance. X-axis stands for dropout rate of both edge and feature, and the y-axis stands for the normalized values.

$\mathbb{E}_{p(v_i^0, v_i^+)} ||f(v_i^0) - f(v_i^+)||^2$ *will increase, i.e.,* $\delta_{aug} \propto \text{GED}(\mathcal{G}^0, \mathcal{G}^+)$. $\text{GED}(\mathcal{G}^0, \mathcal{G}^+)$ *indicates the graph edit distance between* $\mathcal{G}^0$ *and* $\mathcal{G}^+$.

This is a natural assumption that is likely to hold because input with a bigger difference will lead to a bigger difference in output when the model is correctly trained, which is guaranteed by $p(y|v_i^0) = p(y|v_i^+)$. Also we can notice that $\delta_{aug}$ does not only correlates with the magnitude of augmentation, it also implies the alignment performance. So Assumption 2.2 actually means stronger augmentation would lead to larger $\delta_{aug}$ and worse alignment performance which is commonly in empirical experiments as shown in Appendix C.9. This means in graph contrastive learning, augmentation overlap brought by strong augmentation and perfect alignment are mutual exclusive.

With the assumptions, we can get the theorem below:

**Theorem 2.3** (Augmentation and Classification). *If Assumption 2.1 holds, we know that:*

$$\mathbb{E}_{p(v,y)} f(v_y^0)^T \mu_y \geq 1 - \frac{1}{3}\delta_{aug}^2 - \frac{2}{3}\delta_{aug}\delta_{y^+} - \frac{1}{2}\delta_{y^+}^2, \tag{2}$$

$$\mathbb{E}_{p(v,y,y^-)} f(v_y^0)^T \mu_{y^-} \geq 1 - \frac{1}{3}\delta_{aug}^2 - \frac{2}{3}\delta_{aug}\delta_{y^-} - \frac{1}{2}\delta_{y^-}^2, \tag{3}$$

*where* $\mu_y = \mathbb{E}_{p(v,y)}[f(v)]$, $\delta_{y^+}^2 = \mathbb{E}_{p(y,i,j)}||f(v_{y,i}^0) - f(v_{y,j}^0)||^2$ *and* $\delta_{y^-}^2 = \mathbb{E}_{p(y,y^-,i,j)}||f(v_{y,i}^0) - f(v_{y^-,j}^0)||^2$, $y^-$ *stands for a class different from y.*

The proof can be found in Appendix A.1. This shows that the similarity between a node and the center could be roughly represented by the positive pair difference $\delta_{aug}$ and the inter-class/intra-class divergence $\delta_{y^-}, \delta_{y^+}$. We then use positive and negative center similarity to represent $\mathbb{E}_{p(v,y)} f(v_y^0)^T \mu_y$ and $\mathbb{E}_{p(v,y,y^-)} f(v_y^0)^T \mu_{y^-}$, respectively. Note that we calculate the class center $\mu_y$ by averaging nodes from both original view and augmented views, as the class label of nodes after augmentation remains unchanged, our class center should be more precise as more nodes are included.

As we assumed in Assumption 2.2, when the augmentation becomes stronger, positive pair difference *i.e.,* $\delta_{aug}$ would increase, and based on previous researches (Zhu et al., 2020; You et al., 2020; Veličković et al., 2018), the express power of the model would also be enhanced initially, causing intra-class divergence $\delta_{y^+}$ decreasing and inter-class divergence $\delta_{y^-}$ increasing. Therefore, from Inequality (2) we can conclude that when we perform a stronger augmentation, initially, the similarity between node $v_y$ and its center $\mu_y$ (positive center similarity) is hard to predict as $\delta_{aug}$ increases and $\delta_{y^+}$ decreases. However, the right hand side of Inequality (3) should decrease gradually as both $\delta_{aug}$ and $\delta_{y^-}$ increase, so the similarity between node $v_y$ and other center $\mu_{y^-}$ (negative center similarity) would more likely to be lower. In nutshell, with stronger augmentation the negative center similarity is likely to decrease while the positive center similarity may not be increasing.

The experiment shown in Figure 1 confirms our suspicion. We use dropout on edges and features to perform augmentation, and the dropout rate naturally represents the magnitude of augmentation *i.e.,* graph edit distance and present the normalized positive/negative center similarity and downstream accuracy to show the changing tendency. Figure 1 shows that initially, as the dropout rate

increases, positive center similarity may decrease sometimes, but downstream performance could be enhanced as negative center similarity decreases much faster which aligns with our suspicion.

In some datasets, the downstream performance may not be increasing at first, this is mainly because we use 0.05 as the lowest drop rate rather than 0, we show results on lower drop rates and more datasets including shopping graph, graph with heterophily and coauthor network in Appendix C.8. From Figure 1, 11 and 12, we can conclude that contrastive learning mainly contributes to downstream tasks by separating nodes of different classes (decreasing negative center similarity) rather than gathering nodes of the same class (non-increasing positive center similarity), and perfect alignment may not help as it hinders class separating.

We can understand the phenomena intuitively, as the InfoNCE loss $\mathcal{L}_{\text{NCE}}$ can be written as $\mathcal{L}_{\text{NCE}} = \mathbb{E}_{p(v_i^1, v_i^2)} \mathbb{E}_{p(v_i^-)} \left[ -\log \frac{\exp(f(v_i^1)^T f(v_i^2))}{\sum_{\{v_i^-\}} \exp(f(v_i^1)^T f(v_i^-))} \right]$, and the numerator $f(v_i^1)^T f(v_i^2) \propto 1 - \mathbb{E}_{p(v_i)} ||f(v_i^1) - f(v_i^2)|| \propto 1 - \mathbb{E}_{p(v_i)} ||f(v_i^0) - f(v_i^+)|| \propto 1 - \delta_{aug}$, so a higher $\delta_{aug}$ caused by stronger augmentation tends to make the numerator harder to maximize. Then GCL would pay more attention to the minimize the denominator just as shown in Figure 5. And minimizing the denominator is actually separating negative samples which is mainly performing inter-class separating as most negative samples are from the different classes. Thus inter-class separating is enhanced. In contrast, intra-class gathering is weakened due to the existence of same-class samples in the negative set, while the worse alignment performance and better augmentation overlap can hardly help (Wang et al., 2022b; Saunshi et al., 2022). As a result intra-class nodes may not gather closer. Also we can observe from Figure 1 that when we drop too much edges/features, downstream performance decreases sharply, and both positive and negative center similarity increases as too much information is lost and the basic assumption $p(y|v_i^0) = p(y|v_i^+)$ does not hold, then a trivial solution is learned.

## 2.3 AUGMENTATION AND GENERALIZATION

Although GCL with a stronger augmentation may help to improve downstream performance, why it works stays unclear. We need to figure out the relationship between positive pair difference, contrastive loss and downstream performance to further guide algorithm design. We first define the mean cross-entropy (CE) loss below, and use it to represent downstream performance.

**Definition 2.4** (Mean CE loss). *For an encoder $f$ and downstream labels $y \in [1, K]$, we use the mean CE loss $\hat{\mathcal{L}}_{\text{CE}} = \mathbb{E}_{p(v^0, y)} \left[ -\log \frac{\exp(f(v^0)^T \mu_y)}{\sum_{j=1}^{K} \exp(f(v^0)^T \mu_j)} \right]$ to evaluate downstream performance, where $\mu_j = \mathbb{E}_{p(v|y=j)} [f(v)]$.*

It is easy to see that mean CE loss could indicate downstream performance as it requires nodes similar to their respective class center, and different from others class centers. Also it is an upper bound of CE loss $\mathcal{L}_{\text{CE}} = \mathbb{E}_{(v^0, y)} \left[ -\log \frac{\exp(f(v^0)^T \omega_y)}{\sum_{i=1}^{K} \exp(f(v^0)^T \omega_i)} \right]$, where $\omega$ is parameter to train a linear classifier $g(z) = Wz$, $W = [\omega_1, \omega_2, ..., \omega_k]$. Arora et al. (2019) showed that the mean classifier could achieve comparable performance to learned weights, so we analyze the mean CE loss instead of the CE loss in this paper. Similar to Theorem 2.3, we calculate the class center using both original and augmented view nodes, instead of using only the original view nodes (Arora et al., 2019).

**Theorem 2.5** (Generalization and Positive Pair Difference). *If Assumption 2.1 holds, and* ReLU *is applied as activation, then the relationship between downstream performance and* InfoNCE *loss could be represented as:*

$$\hat{\mathcal{L}}_{\text{CE}} \geq \mathcal{L}_{\text{NCE}} - 3\delta_{aug}^2 - 2\delta_{aug} - \log \frac{M}{K} - \frac{1}{2} \text{Var}(f(v^+)|y)$$
$$- \sqrt{\text{Var}(f(v^0)|y)} - e \text{Var}(\mu_y) - O(M^{-\frac{1}{2}}),$$

*where $M$ is number of negative samples[1], $K$ is number of classes, $\text{Var}(f(v^0)|y) = \mathbb{E}_{p(y)} \mathbb{E}_{p(v^0|y)} ||f(v^0) - \mathbb{E}_{p(v|y)} f(v)||^2$ and $\text{Var}(f(v^+)|y) = \mathbb{E}_{p(y)} \mathbb{E}_{p(v^+|y)} ||f(v^+) - \mathbb{E}_{p(v|y)} f(v)||^2$ both mean intra-class variance, and $\text{Var}(\mu_y) = \mathbb{E}_{p(y)} ||\mathbb{E}_{p(v|y)} f(v) - \mathbb{E}_{p(v)} f(v)||^2$ represents the variance of $K$ class centers.*

---

[1]the generalization are correlated with $-\log M - O(M^{-\frac{1}{2}})$, which is decreasing when $M$ increases and $M$ is large, so the theorem encourages large negative samples.

The proof can be found in Appendix A.2. Theorem 2.5 suggests a gap between $\hat{\mathcal{L}}_{\text{CE}}$ and $\mathcal{L}_{\text{NCE}}$, meaning that the encoders that minimize $\mathcal{L}_{\text{NCE}}$ may not yield optimal performance on downstream tasks. Furthermore, it suggests that a higher positive pair difference $\delta_{aug}$ and $\text{Var}(f(v)|y)$ would enhance generalization and potentially improve performance on downstream tasks. Also Inequality (2) also demonstrates that $f(v_y)^T \mu_y \propto [-\text{Var}(f(v)|y)]$ and $f(v_y)^T \mu_y \propto [-\delta_{aug}]$. So better generalization correlates with worse positive center similarity. This aligns with the findings before that better downstream performance may come with a lower positive center similarity.

Theorem 2.5 also highlights the significance of augmentation magnitude in graph contrastive learning algorithms like GRACE (Zhu et al., 2020). A weak augmentation leads to better alignment but also a weak generalization, InfoNCE loss might be relatively low but downstream performance could be terrible (Saunshi et al., 2022). When augmentation gets stronger, although perfect alignment cannot be achieved, it promotes better generalization and potentially leads to improved downstream performance. And when the augmentation is too strong, minimizing the InfoNCE loss becomes challenging (Li et al., 2022), leading to poorer downstream performance. Therefore, it is crucial to determine the magnitude of augmentation and how to perform augmentation as it directly affects contrastive performance and generalization.

## 3 FINDING BETTER AUGMENTATION

Previous sections have revealed that perfect alignment, which minimizes the positive pair difference $\delta_{aug}$ to 0 may not help downstream performance. Instead a stronger augmentation that leads to larger $\delta_{aug}$ will benefit generalization while weakening contrastive learning process. Therefore, we need to find out how to perform augmentation to strike a better balance between positive pair difference and contrastive loss, leading to better downstream performance.

### 3.1 INFORMATION THEORY PERSPECTIVE

As shown by Oord et al. (2018), $\mathcal{L}_{\text{NCE}}$ is actually a lower bound of mutual information. Additionally, $\text{Var}(f(v^0)|y)$, $\text{Var}(f(v^+|y))$ and $\text{Var}(\mu_y)$ can be represented by inherent properties of the graph and the positive pair difference $\delta_{aug}$. Thus, Theorem 2.5 could be reformulated as follows:

**Corollary 3.1** (CE with Mutual Information). *If Assumption 2.1 holds, the relationship between downstream performance, mutual information between views and positive pair difference could be represented as:*

$$\hat{\mathcal{L}}_{\text{CE}} \geq \log(K) - I(f(v^1), f(v^2)) - g(\delta_{aug}) - O(M^{-\frac{1}{2}}), \tag{4}$$

*where $I(f(v^1), f(v^2))$ stands for the mutual information between $f(v^1)$ and $f(v^2)$, $g(\delta_{aug})$ is monotonically increasing, and is defined in Appendix A.3.*

The proof can be found in Appendix A.3. Corollary 3.1 suggests that the best augmentation would be one that maximize the mutual information and positive pair difference. To verify this, we propose a simple but effective method. We recognize important nodes, features and edges, then leave them unchanged during augmentation to increase mutual information. Then for those unimportant ones, we should perform stronger augmentation to increase the positive pair difference.

Similar to Wei et al. (2023), we utilize gradients to identify which feature of node $v$ is relatively important and carries more information. We calculate the importance of feature by averaging the feature importance across all nodes, the importance of node $v$ could be calculated by simply averaging the importance of its features, and then use the average of the two endpoints to represent the importance of an edge:

$$\alpha_{v,p} = \frac{\partial \mathcal{L}_{\text{NCE}}}{\partial x_{v,p}}, \quad \alpha_p = \text{ReLU}\left(\frac{1}{|V'|}\sum_v \alpha_{v,p}\right),$$

$$\alpha_v = \text{ReLU}\left(\frac{1}{|P'|}\sum_p \alpha_{v,p}\right), \quad \alpha_{e_{i,j}} = \left(\alpha_{v_i} + \alpha_{v_j}\right)/2,$$

where $\alpha_{v,p}$ means importance of the $p^{th}$ feature of node $v$, $\alpha_p$ means the importance of $p^{th}$ feature, $\alpha_v$ means importance of node $v$, and $\alpha_{e_{i,j}}$ means the importance of edge $(v_i, v_j)$.

For those edges/features with high importance, we should keep them steady and do no modification during augmentation. For those with relatively low importance, we can freely mask those edges/features, but we should make sure that the number of masked edges/features is greater than the number of kept ones to prevent $\delta_{aug}$ from decreasing. The process can be described by the following equation:

$$\tilde{\boldsymbol{A}} = \boldsymbol{A} * (\boldsymbol{M}_e \vee \boldsymbol{S}_e \wedge \boldsymbol{D}_e), \quad \tilde{\boldsymbol{F}} = \boldsymbol{F} * (\boldsymbol{M}_f \vee \boldsymbol{S}_f \wedge \boldsymbol{D}_f),$$

where $*$ is hadamard product, $\vee$ stands for logical OR, $\wedge$ stands for logical AND. $\boldsymbol{M}_e$, $\boldsymbol{M}_f$ represent the random mask matrix, which could be generated using any mask method, $\boldsymbol{S}_e$, $\boldsymbol{S}_f$ are the importance based retain matrix, it tells which edge/feature is of high importance and should be retained. For the top $\xi$ important edges/features, we set $\boldsymbol{S}_e$, $\boldsymbol{S}_f$ to 1 with a probability of 50% and to 0 otherwise. $\boldsymbol{D}_e$, $\boldsymbol{D}_f$ show those edges/features should be deleted to increase $\delta_{aug}$, for the least $2\xi$ important edges/features, we also set $\boldsymbol{D}_e$, $\boldsymbol{D}_f$ to 0 with a probability of 50% and to 1 otherwise. It is worth noting that $\delta_{aug}$ is defined as $\mathbb{E}_{p(v_i^0, v_i^+)}||f(v_i^+) - f(v_i^0)||$ rather than $\mathbb{E}_{p(v_i^1, v_i^2)}||f(v_i^1) - f(v_i^2)||$, therefore, we applied this deletion on both views.

This is a simple method, and the way to measure importance can be replaced by any other methods. It can be easily integrated into any other graph contrastive learning methods that require edge/feature augmentation. There are many details that could be optimized, such as how to choose which edges/features to delete and the number of deletions. However, since this algorithm is primarily intended for theoretical verification, we just randomly select edges to be deleted and set the number to twice the number of edges kept.

In fact, most graph contrastive learning methods follow a similar framework to maximize mutual information and implicitly increase positive pair difference as discussed in Appendix B.1.

## 3.2 GRAPH SPECTRUM PERSPECTIVE

In this section, we attempt to analyze InfoNCE loss and positive pair difference from graph spectrum perspective. We start by representing them using the spectrum of the adjacency matrix $\boldsymbol{A}$.

**Theorem 3.2** (Theorem 1 of Liu et al. (2022) Restated). *Given adjacency martix $\boldsymbol{A}$ and the generated augmentation $\boldsymbol{A}'$, $\boldsymbol{A}''$, the $i^{th}$ eigenvalues of $\boldsymbol{A}'$ and $\boldsymbol{A}''$ are $\lambda_i'$, $\lambda_i''$, respectively. The following upper bound is established:*

$$\mathcal{L}_{\text{NCE}} \geq N \log N - (N+1) \sum_i \theta_i \lambda_i' \lambda_i'', \tag{5}$$

*where $\theta_i$ is the adaptive weight of the $i^{th}$ term, the detail of $\theta_i$ is discussed in* Appendix C.4.

**Corollary 3.3** (Spectral Representation of $\delta_{aug}$). *If Assumption 2.1 holds, and $\lambda_i'$, $\lambda_i''$ are $i^{th}$ eigenvalues of $\boldsymbol{A}'$ and $\boldsymbol{A}''$, respectively, then:*

$$2\delta_{aug} \geq \mathbb{E}_{p(v_i^1, v_i^2)}||f(v_i^1) - f(v_i^2)|| \geq \sqrt{2 - \frac{2}{N} \sum_i \theta_i \lambda_i' \lambda_i''}. \tag{6}$$

Theorem 2.5 suggests that we should strive to make $\mathcal{L}_{\text{NCE}}$ small while increase $\delta_{aug}$, but they are kindly mutually exclusive. As shown in Theorem 3.2, and Corallary 3.3 proved in Appendix A.4, when $\theta_i$ is positive, a small $\mathcal{L}_{\text{NCE}}$ requires for large $|\lambda_i|$ while a large $\delta_{aug}$ requires for small $|\lambda_i|$, and it works exclusively too when $\theta_i$ is negative. As contrastive learning is trained to minimize $\mathcal{L}_{\text{NCE}}$, $\theta$s are going to increase as the training goes, so we can assume that $\theta$s will be positive, the detailed discussion and exact definition of $\theta$ can be found in Appendix C.4. Since $\theta$s are trained parameters that we have limited control over, we turn to adjusting $\lambda$s through data augmentation. Therefore, to achieve a better trade-off, we should decrease $|\lambda_i|$ while keep InfoNCE loss also decreasing.

In fact, reducing $|\lambda_i|$ actually reduces the positive $\lambda_i$ and increases the negative $\lambda_i$, which is trying to smoothen the graph spectrum and narrow the gap between the spectrum. As suggested by Yang et al. (2022), graph convolution operation with an unsmooth spectrum results in signals correlated to the eigenvectors corresponding to larger magnitude eigenvalues and orthogonal to the eigenvectors corresponding to smaller magnitude eigenvalues. So if $|\lambda_i| \gg |\lambda_j|$, then $\text{sim}(f(v), e_i) \gg \text{sim}(f(v), e_j)$, where $e_i$ denotes the eigenvector corresponding to $\lambda_i$, causing all

representations similar to $e_i$. Therefore, an unsmooth spectrum may lead to similar representations. This can also be observed from Inequality (6), where a higher $|\lambda_i|$ will reduce the positive pair difference, making $f(v_i^1)$ and $f(v_i^2)$ more similar.

We now know that smoothing the graph spectrum can help with graph contrastive learning. The question is how to appropriately smooth the spectrum. We propose a simple method. As the training aims to minimize $\mathcal{L}_{\text{NCE}}$, the parameter $\theta_i$s are supposed to increase. Therefore, we can use $\theta_i$ as a symbol to show whether the model is correctly trained. When $\theta_i$ gradually increases, we can decrease $\lambda$ as needed. However, when $\theta_i$ starts to decrease, it is likely that the change on the spectrum is too drastic, and we should take a step back. The process could be described as follows:

$$\lambda_i = \lambda_i + \text{direction}_i * \lambda_i * \alpha, \quad \text{direction}_i = \begin{cases} -1, & \text{cur}(\theta_i) - \text{pre}(\theta_i) \geq \epsilon \\ 1, & \text{cur}(\theta_i) - \text{pre}(\theta_i) \leq -\epsilon, \\ 0, & \text{otherwise} \end{cases}$$

where $\alpha$ is a hyperparameter that determines how much we should decrease/increase $\lambda_i$. $\epsilon$ is used to determine whether $\theta_i$ is increasing, decreasing, or just staying steady. $\text{cur}(\theta_i)$ and $\text{pre}(\theta_i)$ represents the current and previous $\theta_i$ respectively.

In this way, the contrastive training will increase $\theta_i$ and result in a lower $\mathcal{L}_{\text{NCE}}$, while we justify $\lambda_i$ to achieve a better positive pair difference, which promises a better generalization ability. However, just like Section 3.1, the method is quite simple, in fact this method is more a data preprocessing rather than augmentation, but it is capable of verifying the theory and guide algorithm design, as we decrease $\lambda'$ by directly decreasing $\lambda$, but it could also be achieved by augmentation methods.

Also some spectral augmentations implicitly decreases $|\lambda|$s as shown in Appendix B.2.

## 4  EXPERIMENTS

In this section, we mainly evaluate the performance of the methods we proposed on six datasets: Cora, CiteSeer, PubMed, DBLP, Amazon-Photo and Amazon-Computer. We select 3 contrastive learning GNN, GRACE (Zhu et al., 2020), GCA (Zhu et al., 2021), AD-GCL (Suresh et al., 2021), and integrate those models with our proposed methods to verify its applicability and correctness of the theory. Details of datasets and baselines are in Appendix C.1. The results are summarized in Table 1. We further investigate the positive/negative center similarity in Appendix C.6, the hyperparameter sensitivity is studied in Appendix C.7, and the change of $\theta$ and the spectrum while training is shown in Appendix C.5.

Table 1: Quantitative results on node classification, algorithm+I stands for the algorithm with information augmentation, and algorithm+S stands for the algirithm with spectrum augmentation

| Datasets | Cora | | CiteSeer | | PubMed | | DBLP | | Amazon-P | | Amazon-C | | pvalue |
|---|---|---|---|---|---|---|---|---|---|---|---|---|---|
| | Mi-Fi | Ma-Fi | Mi-Fi | Ma-Fi | Mi-Fi | Ma-Fi | Mi-Fi | Ma-Fi | Mi-Fi | Ma-Fi | Mi-Fi | Ma-Fi | |
| GCN | 83.31 | 81.97 | 69.81 | 66.44 | 85.36 | 84.88 | 81.26 | 75.40 | 93.28 | 91.78 | 88.11 | 81.57 | |
| GAT | 83.83 | 82.45 | 70.31 | 66.76 | 84.04 | 83.43 | 81.92 | 75.87 | 93.17 | 91.84 | 86.82 | 78.37 | |
| SpCo | 83.45 | 82.16 | 69.90 | 66.79 | OOM | OOM | 83.61 | 79.25 | 91.56 | 89.85 | 83.37 | 80.14 | |
| GCS | 83.39 | 82.11 | 68.73 | 67.92 | 84.92 | 83.70 | 83.38 | 78.82 | 90.15 | 89.21 | 86.54 | 84.75 | |
| GRACE | 82.52 | 81.23 | 68.44 | 63.73 | 84.97 | 84.51 | 84.01 | 79.63 | 91.17 | 89.09 | 86.36 | 84.15 | |
| GRACE+I | **83.33** | **82.23** | **70.47** | 64.83 | 84.99 | 84.57 | 84.39 | 80.24 | 91.13 | 89.11 | **86.61** | **84.77** | 0.155 |
| GRACE+S | 83.25 | 81.85 | 69.87 | **64.92** | **85.03** | **84.62** | **84.47** | **80.33** | **91.91** | **90.09** | 86.61 | 84.66 | 0.003 |
| GCA | 83.74 | 82.28 | 71.09 | 66.43 | 85.38 | **85.07** | 83.99 | 79.82 | 91.67 | 90.21 | 86.77 | 85.18 | |
| GCA+I | **84.71** | **83.42** | 71.24 | **67.23** | 85.38 | 84.89 | 84.29 | 79.91 | 91.94 | **90.40** | 86.60 | 84.12 | 0.089 |
| GCA+S | 83.51 | 82.30 | 70.95 | 65.31 | 85.28 | 84.98 | **84.49** | **80.28** | **92.02** | 90.36 | **86.97** | **85.30** | 0.147 |
| AD GCL | 81.68 | 79.83 | 70.01 | 64.17 | 84.77 | 84.29 | 83.14 | 78.86 | 91.34 | 89.28 | 84.80 | 82.04 | |
| AD GCL+I | **83.06** | 81.20 | 71.06 | **64.69** | **85.52** | **85.00** | **83.51** | 79.05 | **91.91** | **90.24** | **86.02** | **84.12** | 0.03 |
| AD GCL+S | 82.96 | **81.39** | **71.35** | 63.88 | 85.08 | 84.60 | 83.45 | **79.13** | 91.79 | 89.94 | 85.49 | 82.52 | 0.06 |

From Table 1, we can observe that GRACE+I (GRACE with information augmentation) and GRACE+S (GRACE with spectrum augmentation) both improve the downstream performance. This improvement is significant for GRACE since it primarily performs random dropout, resulting in the loss of valuable information. But for GCA, the information augmentation only brings minor improvements. This is because GCA already drops the unimportant ones with a higher probability, allowing it to capture sufficient information, especially on large graphs. AD-GCL aggressively drops

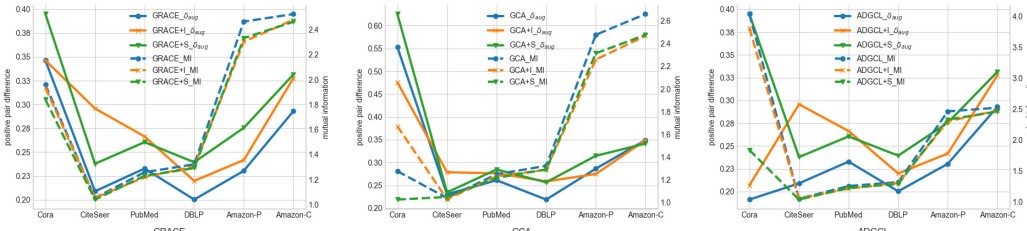

Figure 2: Positive pair difference and InfoNCE, GRACE+I stands for GRACE with information augmentation, and GRACE+S stands for GRACE with spectrum augmentation. GRACE+x_MI means mutual information between two views after training, and GRACE+x_$\delta_{aug}$ is positive pair difference caused by the method.

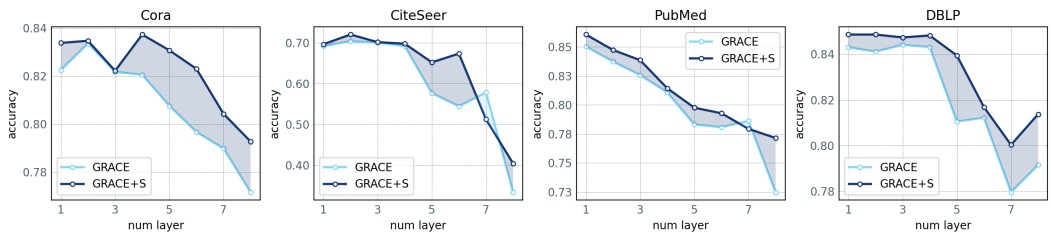

Figure 3: Accuracy on downstream tasks with different number of layers. GRACE is the original algorithm (Zhu et al., 2020), and GRACE+S stands for GRACE with spectrum augmentation.

as much information as possible to eliminate irrelevant information while some important ones are also dropped, so the information augmentation helps greatly. Overall, our methods improve the performance of original algorithm and helps downstream tasks, the p-value on the averaged performance are also shown in Table 1.

### 4.1 POSITIVE PAIR DIFFERENCE

Figure 2 shows that for all three algorithms, our methods capture similar information while achieving a larger positive pair difference. This indicates that we keep the contrastive loss while enhancing its generalization, resulting in improved downstream performance. Similar to the result of Table 1, the improvement of $\delta_{aug}$ on GRACE and AD-GCL are much sharper as GCA has already tried to increase positive pair difference and achieve a balance with the InfoNCE loss as discussed in Appendix B.1.

### 4.2 OVER-SMOOTH

While reducing $|\lambda_i|$, we obtain a graph with smoother spectrum, and could relieve the over-smooth. This, in turn, enables the application of relatively more complex models. We can verify this by simply stacking more layers. As shown in Figure 3, if applied spectrum augmentation, the model tends to outperform the original algorithm especially with more layer, and the best performance may come with a larger number of layers, which indicates that more complicated models could be applied and our method successfully relieve over-smooth.

## 5 CONCLUSION

In this paper, we investigate the impact of contrastive learning on downstream tasks and propose that perfect alignment does not necessarily lead to improved performance. Instead, we find that a relatively large positive pair difference is more beneficial for generalization. Building upon this insight, we introduce two simple yet effective methods to strike a balance between the contrastive loss and positive pair difference.

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

## A  THEORETICAL PROOF

### A.1  PROOF OF LEMMA 2.3

If we set $\delta_{y+}^2 = \mathbb{E}_{p(y,i,j)}||f(v_{y,i}^0) - f(v_{y,j}^0)||^2$, and $\delta_{y+}^2 = \mathbb{E}_{p(y,y',i,j)}||f(v_{y,i}^0) - f(v_{y',j}^0)||^2$. Then with Assumption 2.1 and jensen inequality, we know that $\mathbb{E}_{p(v_i)}||f(v_i^0) - f(v_i^+)||^2 \leq \delta_{aug}^2$, $\mathbb{E}_{p(v_i)}||f(v_i^0) - f(v_i^+)|| \leq \delta_{aug}$ and $\mathbb{E}_{p(y,i,j)}||f(v_{y,i}^0) - f(v_{y,j}^0)|| \leq \delta_{y+}$, $\mathbb{E}_{p(y,y',i,j)}||f(v_{y,i}^0) - f(v_{y',j}^0)|| \leq \delta_{y-}$. Therefore, we can get the inequality below:

$$
\begin{aligned}
\mathbb{E}_{p(v_{y,i},v_{y,j}|y)}||f(v_{y,i}^+) - f(v_{y,j}^0)||^2 &\leq \mathbb{E}_{p(v_{y,i},v_{y,j}|y)}||f(v_{y,i}^+) - f(v_{y,i}^0)||^2 + \mathbb{E}_{p(v_{y,i},v_{y,j}|y)}||f(v_{y,i}^0) - f(v_{y,j}^0)||^2 \\
&\quad + 2\mathbb{E}_{p(v_{y,i},v_{y,j}|y)}||f(v_{y,i}^+) - f(v_{y,j}^0)|| \cdot ||f(v_{y,i}^0) - f(v_{y,j}^0)|| \\
&\leq \delta_{aug}^2 + \delta_{y+}^2 + 2\delta_{aug}\delta_{y+} \\
&= (\delta_{aug} + \delta_{y+})^2.
\end{aligned}
$$

As $\mu_y = \mathbb{E}_{p(v_y|y)}[f(v_y)] = \frac{1}{3}\mathbb{E}_{p(v_y^0|y)}f(v_y^0) + \frac{2}{3}\mathbb{E}_{p(v_y^+|y)}f(v_y^0)$, we know that,

$$
\begin{aligned}
\mathbb{E}_{p(v_y^{0'}|y)}f(v_y^{0'})^T\mu_y &= \mathbb{E}_{p(v_y^{0'}|y)}f(v_y^{0'})^T(\frac{1}{3}\mathbb{E}_{p(v_y^0|y)}f(v_y^0) + \frac{2}{3}\mathbb{E}_{p(v_y^+|y)}f(v_y^0)) \\
&= \frac{1}{3}\mathbb{E}_{p(v_y^{0'}|y)}\mathbb{E}_{p(v_y^0|y)}f(v_y^{0'})^Tf(v_y^0) + \frac{2}{3}\mathbb{E}_{p(v_y^{0'}|y)}\mathbb{E}_{p(v_y^+|y)}f(v_y^{0'})^Tf(v_y^+).
\end{aligned}
$$

assume that $\mathbb{E}_{p(a,b)}||a - b||^2 \leq c^2$, $||a|| = ||b|| = 1$, then

$$
\begin{aligned}
\mathbb{E}_{p(a,b)}(a^T - b^T)(a - b) &\leq c^2 \\
\mathbb{E}_{p(a,b)}[a^Ta - a^Tb - b^Ta + b^Tb] &\leq c^2 \\
\mathbb{E}_{p(a,b)}[2 - 2a^Tb] &\leq c^2 \\
\mathbb{E}_{p(a,b)}a^Tb &\geq \frac{2 - c^2}{2} = 1 - \frac{c^2}{2}.
\end{aligned}
$$

As we already know that $\mathbb{E}_{p(y,y',i,j)}||f(v_{y,i}^0) - f(v_{y',j}^0)||^2 \leq \delta_{y+}^2$ and $\mathbb{E}_{p(v_{y,i},v_{y,j}|y)}||f(v_{y,i}^+) - f(v_{y,j}^0)||^2 \leq (\delta_{aug} + \delta_{y+})^2$. So $\mathbb{E}_{p(v_y^{0'}|y)}\mathbb{E}_{p(v_y^0|y)}f(v_y^{0'})^Tf(v_y^0) \geq 1 - \frac{\delta_{y+}^2}{2}$ and $\mathbb{E}_{p(v_y^{0'}|y)}\mathbb{E}_{p(v_y^+|y)}f(v_y^{0'})^Tf(v_y^+) \geq 1 - \frac{(\delta_{aug}+\delta_{y+})^2}{2}$.

Then, we can calculate $\mathbb{E}_{p(v_y^{0'}|y)}f(v_y^{0'})^T\mu_y$ as below:

$$
\begin{aligned}
\mathbb{E}_{p(v_y^{0'}|y)}f(v_y^{0'})^T\mu_y &= \frac{1}{3}\mathbb{E}_{p(v_y^{0'}|y)}\mathbb{E}_{p(v_y^0|y)}f(v_y^{0'})^Tf(v_y^0) + \frac{2}{3}\mathbb{E}_{p(v_y^{0'}|y)}\mathbb{E}_{p(v_y^+|y)}f(v_y^{0'})^Tf(v_y^+) \\
&\geq 1 - \frac{\delta_{aug}^2}{3} - \frac{2\delta_{aug}\delta_{y+}}{3} - \frac{\delta_{y+}^2}{2}.
\end{aligned}
\tag{7}
$$

Similarly, we know that $\mathbb{E}_{p(v_y^0,y^-|y)}f(v_y^0)^T\mu_{y^-} \geq 1 - \frac{\delta_{aug}^2}{3} - \frac{2\delta_{aug}\delta_{y-}}{3} - \frac{\delta_{y-}^2}{2}$.

## A.2 Proof of Theorem 2.5

$$\hat{\mathcal{L}}_{\text{CE}} = \underbrace{-\mathbb{E}_{p(v_i^0,y)}f(v_i^0)^T\mu_y}_{\Lambda_1} + \underbrace{\mathbb{E}_{p(v_i^0)}\log\sum_{i=j}^{K}\exp(f(v_i^0)^T\mu_j)}_{\Lambda_2}.$$

$$\begin{aligned}
\Lambda_1 &= -\mathbb{E}_{p(v_i^0,y)}f(v_i^0)^T\mu_y \\
&= -\mathbb{E}_{p(v_i^0,y)}\left[f(v_i^0)^Tf(v_i^+) + f(v_i^0)^T(\mu_y - f(v_i^+))\right] \\
&\overset{(a)}{\geq} -\mathbb{E}_{p(v_i^0,v_i^+,y)}f(v_i^0)^Tf(v_i^+) - \mathbb{E}_{p(v_i^+,y)}||f(v_i^+) - \mu_y|| \\
&\geq -\mathbb{E}_{p(v_i^0,v_i^+,y)}f(v_i^0)^Tf(v_i^+) - \mathbb{E}_{p(v_i^0,v_i^+,y)}||f(v_y^+) - f(v_y^0)|| - \mathbb{E}_{p(v_i^0,v_i^+,y)}||f(v_y^0) - \mu_y|| \\
&\overset{(b)}{\geq} -\mathbb{E}_{p(v_i^0,v_i^+,y)}f(v_i^1)^Tf(v_i^2) - 3\delta_{aug}^2 - \delta_{aug} - \mathbb{E}_{p(v_i^0,v_i^+,y)}||f(v_y^0) - \mu_y||.
\end{aligned}$$

(a) $f(v_i^0)^T(\mu_y - f(v_i^+)) \leq (\frac{\mu_y - f(v_i^+)}{||\mu_y - f(v_i^+)||})^T(\mu_y - f(v_i^+)) = ||\mu_y - f(v_i^+)||.$

(b) $\mathbb{E}_{p(v_i^0,v_i^+)}||f(v_i^0) - f(v_i^+)||^2 \leq \delta_{aug}^2$, then:

$$\begin{aligned}
\delta_{aug}^2 &\geq \mathbb{E}_{p(v_i^0,v_i^1)}(f(v_i^0) - f(v_i^1))^T \cdot (f(v_i^0) - f(v_i^1)) \\
&= \mathbb{E}_{p(v_i^0,v_i^1,v_i^2)}(f(v_i^0) - f(v_i^1))^T \cdot (f(v_i^0) - f(v_i^1) + f(v_i^2) - f(v_i^2)) \\
&= \mathbb{E}_{p(v_i^0,v_i^1,v_i^2)}f(v_i^0)^Tf(v_i^0) - f(v_i^0)^Tf(v_i^1) + f(v_i^0)^Tf(v_i^2) - f(v_i^0)^Tf(v_i^2) \\
&\quad - f(v_i^1)^Tf(v_i^0) + f(v_i^1)^Tf(v_i^1) - f(v_i^1)^Tf(v_i^2) + f(v_i^1)^Tf(v_i^2) \\
&= 2 + \mathbb{E}_{p(v_i^0,v_i^1,v_i^2)}\left[-2f(v_i^0)^Tf(v_i^1) + f(v_i^0)^Tf(v_i^2) - f(v_i^0)^Tf(v_i^2) - f(v_i^1)^Tf(v_i^2) + f(v_i^1)^Tf(v_i^2)\right] \\
&\overset{(c)}{\geq} 2 - 2 + \mathbb{E}_{p(v_i^0,v_i^1,v_i^2)}\left[f(v_i^0)^Tf(v_i^2) - 1 - f(v_i^1)^Tf(v_i^2) + 1 - 2\delta_{aug}^2\right] \\
&= \mathbb{E}_{p(v_i^0,v_i^1,v_i^2)}\left[f(v_i^0)^Tf(v_i^2) - f(v_i^1)^Tf(v_i^2)\right] - 2\delta_{aug}^2.
\end{aligned}$$

So, we can get the relationship between $\mathbb{E}_{p(v_i^0,v_i^1,v_i^2)}f(v_i^0)^Tf(v_i^2)$ and $\mathbb{E}_{p(v_i^0,v_i^1,v_i^2)}f(v_i^1)^Tf(v_i^2) - 2\delta_{aug}^2$ as below:

$$\delta_{aug}^2 \geq \mathbb{E}_{p(v_i^0,v_i^1,v_i^2)}f(v_i^0)^Tf(v_i^2) - \mathbb{E}_{p(v_i^0,v_i^1,v_i^2)}f(v_i^1)^Tf(v_i^2) - 2\delta_{aug}^2,$$
$$\mathbb{E}_{p(v_i^0,v_i^1,v_i^2)}f(v_i^0)^Tf(v_i^2) \leq \mathbb{E}_{p(v_i^0,v_i^1,v_i^2)}f(v_i^1)^Tf(v_i^2) + 3\delta_{aug}^2.$$

As $v_i^2$ is an augmented node, we can get that,

$$\mathbb{E}_{p(v_i^0,v_i^+)}f(v_i^0)^Tf(v_i^+) \leq \mathbb{E}_{p(v_i^0,v_i^1,v_i^2)}f(v_i^1)^Tf(v_i^2) + 3\delta_{aug}^2.$$

(c) $f(v_i^0)^Tf(v_i^1) \leq 1$, $f(v_i^0)^Tf(v_i^2) \leq 1$, and $\mathbb{E}_{p(v_i^1,v_i^2)}f(v_i^1)^Tf(v_i^2) \geq \frac{2 - \mathbb{E}_{p(v_i^1,v_i^2)}||f(v_i^1) - f(v_i^2)||^2}{2} \geq 1 - \frac{\mathbb{E}_{p(v_i^1,v_i^2)}(||f(v_i^1) - f(v_i^0)|| + ||f(v_i^0) - f(v_i^2)||)^2}{2} \geq 1 - 2\delta_{aug}^2.$

**Lemma A.1** ((Budimir et al., 2000) Corollary 3.5 (restated)). *Let $g : \mathbb{R}^m \to \mathbb{R}$ be a differentiable convex mapping and $z \in \mathbb{R}^n$. Suppose that $g$ is $L$- smooth with the constant $L > 0$, i.e. $\forall x, y \in \mathcal{R}^m, ||\nabla g(x) - \nabla g(y)|| \leq L||x - y||$. Then we have*

$$0 \leq \mathbb{E}_{p(z)}g(z) - g\left(\mathbb{E}_{p(z)}z\right) \leq L\left[\mathbb{E}_{p(z)}||z||^2 - ||\mathbb{E}_{p(z)}z||^2\right] = L\sum_{j=1}^{n}\text{Var}(z^{(j)}),$$

*where $z^{(j)}$ denotes the $j$-th dimension of $v$.*

**Lemma A.2** ((Wang et al., 2022b) Lemma A.2. restated). *For* LSE $:= \log \mathbb{E}_{p(z)} \exp(f(v)^\top g(z))$, *we denote its (biased) Monte Carlo estimate with $M$ random samples $z_i \sim p(z), i = 1, \ldots, M$ as $\widehat{\text{LSE}}_M = \log \frac{1}{M} \sum_{i=1}^{M} \exp(f(v)^\top g(z_i))$. Then the approximation error $A(M)$ can be upper bounded in expectation as*

$$A(M) := \mathbb{E}_{p(v, z_i)} |\widehat{\text{LSE}}(M) - \text{LSE}| \leq \mathcal{O}(M^{-1/2}).$$

*We can see that the approximation error converges to zero in the order of $M^{-1/2}$.*

$$\Lambda_2 = \mathbb{E}_{p(v_i^0)} \log \sum_{j=1}^{K} \exp(f(v_i^0)^T \mu_{y_j})$$

$$= \mathbb{E}_{p(v_i^0)} \log \frac{1}{K} \sum_{i=j}^{K} \exp(f(v_i^0)^T \mu_{y_j}) + \log K$$

$$= \mathbb{E}_{p(v_i^0)} \log \mathbb{E}_{p(y_j)} \exp(f(v_i^0)^T \mu_{y_j}) + \log K$$

$$\overset{(d)}{\geq} \mathbb{E}_{p(v_i^1)} \log \mathbb{E}_{p(y_j)} \exp(f(v_i^1)^T \mu_{y_j}) - \delta_{aug} - e \sum_{j=1}^{n} \text{Var}(\mu_j) + \log K$$

$$\overset{(e)}{\geq} \mathbb{E}_{p(v_i^1)} \mathbb{E}_{p(y_i)} \log \frac{1}{M} \sum_{j=1}^{M} \exp(f(v_i^1)^T \mu_{y_j}) - A(M) + \log K - \delta_{aug} - e \sum_{j=1}^{n} \text{Var}(\mu_j)$$

$$= \mathbb{E}_{p(v_i^1)} \mathbb{E}_{p(y_i)} \log \frac{1}{M} \sum_{j=1}^{M} \exp(\mathbb{E}_{p(v_i^-|y_i^-)} f(v_i^1)^T f(v_i^-)) - A(M) + \log K - \delta_{aug} - e \sum_{j=1}^{n} \text{Var}(\mu_j)$$

$$\overset{(f)}{\geq} \mathbb{E}_{p(v_i^1)} \mathbb{E}_{p(y_i)} \mathbb{E}_{p(v_i^-|y_i^-)} \log \frac{1}{M} \sum_{i=1}^{M} \exp(f(v_i^1)^T f(v_i^-))$$

$$- \frac{1}{2} \sum_{j=1}^{m} \text{Var}(f_j(v^-|y)) - A(M) + \log K - \delta_{aug} - e \sum_{j=1}^{n} \text{Var}(\mu_j)$$

$$= \mathbb{E}_{p(v_i^1)} \mathbb{E}_{p(y_i)} \mathbb{E}_{p(v_i^-|y_i^-)} \log \sum_{i=1}^{M} \exp(f(v_i^1)^T f(v_i^-))$$

$$- \log M - \frac{1}{2} \sum_{j=1}^{m} \text{Var}(f_j(v^-|y)) - A(M) + \log K - \delta_{aug} - e \sum_{j=1}^{n} \text{Var}(\mu_j).$$

(d) We can show that: $\exp([f(v)^T \mu_{y_j}]$ is convex, and $u_{y_j}$ satisfy e-smooth,

$$||\frac{\partial \exp(f(v)^T a)}{\partial a} - \frac{\partial \exp(f(v)^T b)}{\partial b}||$$

$$= || \exp(f(v)^T a) f(v) - \exp(f(v)^T b) f(v))||$$

$$= | \exp(f(v)^T a) - \exp(f(v)^T b)| \cdot ||f(v)||$$

$$\leq | \exp(f(v)^T a) - \exp(f(v)^T b)|$$

$$\leq e||(f(v)^T)(a - b)|| \quad (f(v)^T a, f(v)^T b \leq 1, \text{ so the biggest slope is } e)$$

$$\leq e||a - b||.$$

So according to Lemma A.1, we get,

$$\mathbb{E}_{p(y_j)} \exp([f(v_i^1)^T \mu_{y_j}]) \leq \exp([f(v_i^1)^T \mathbb{E}_{p(y_j)} \mu_{y_j}]) + e \sum_{j=1}^{n} \text{Var}(\mu_j)$$

$$= \exp(f(v_i^1)^T \mu) + e \sum_{j=1}^{n} \text{Var}(\mu_j).$$

Then, we can calculate the difference between $\log \mathbb{E}_{p(y_j)} \exp([f(v_i^0)^T \mu_{y_j}])$ and $\log \mathbb{E}_{p(y_j)} \exp([f(v_i^1)^T \mu_{y_j}])$ by applying reversed Jensen and Jensen inequality, respectively.

$$\log \mathbb{E}_{p(y_j)} \exp([f(v_i^1)^T \mu_{y_j}]) - \log \mathbb{E}_{p(y_j)} \exp([f(v_i^0)^T \mu_{y_j}])$$

$$\leq \log \mathbb{E}_{p(y_j)} \exp([f(v_i^1)^T \mu_{y_j}]) - [f(v_i^0)^T \mu]$$

$$\leq \log \left[ \exp(f(v_i^1)^T \mu) + e \sum_{j=1}^n \mathrm{Var}(\mu_j) \right] - [f(v_i^0)^T \mu]$$

$$= \log \left[ \exp(f(v_i^1)^T \mu) \right] + \log \left[ 1 + \frac{e \sum_{j=1}^n \mathrm{Var}(\mu_j)}{\exp(f(v_i^1)^T \mu)} \right] - [f(v_i^0)^T \mu]$$

$$\leq f(v_i^1)^T \mu - f(v_i^0)^T \mu + \log \left[ 1 + e \sum_{j=1}^n \mathrm{Var}(\mu_j) \right] \quad (e^2 \sum_{j=1}^n \mathrm{Var}(\mu_j), \text{ if not ReLU})$$

$$\leq (f(v_i^1)^T - f(v_i^0)^T)\mu + e \sum_{j=1}^n \mathrm{Var}(\mu_j)$$

$$\leq (f(v_i^1) - f(v_i^0))^T \frac{||\mu||}{||f(v_i^1) - f(v_i^0)||} (f(v_i^1) - f(v_i^0)) + e \sum_{j=1}^n \mathrm{Var}(\mu_j)$$

$$\leq (f(v_i^1) - f(v_i^0))^T \frac{1}{||f(v_i^1) - f(v_i^0)||} (f(v_i^1) - f(v_i^0)) + e \sum_{j=1}^n \mathrm{Var}(\mu_j)$$

$$\leq \delta_{aug} + e \sum_{j=1}^n \mathrm{Var}(\mu_j).$$

(e) We adopt a Monte Carlo estimation with $M$ samples from $p(y)$ and bound the approximation error with Lemma A.2.

(f) We also uses Lemma A.1, and as proof by Wang et al. (2022b), logsumexp is L-smooth for $L = \frac{1}{2}$.

$$\mathcal{L}_{\mathrm{CE}} = \Lambda_1 + \Lambda_2$$

$$\geq -\mathbb{E}_{p(v,y)} f(v_i^1)^T f(v_i^2) - 3\delta_{aug}^2 - \delta_{aug} - \mathbb{E}_{p(v^0,y)} ||f(v_y^0) - \mu_y||$$

$$\quad + \mathbb{E}_{p(v_i^1)} \mathbb{E}_{p(y_i)} \mathbb{E}_{p(v_i^-|y_i)} \log \sum_{i=1}^M \exp(f(v_i^1)^T f(v_i^-))$$

$$\quad - \log M - \frac{1}{2} \sum_{j=1}^m \mathrm{Var}(f_j(v^-|y)) - A(M) + \log K - \delta_{aug} - e \sum_{j=1}^n \mathrm{Var}(\mu_j)$$

$$= \left[ -\mathbb{E}_{p(v_i^1,v^2)} f(v_i^1)^T f(v_i^2) + \mathbb{E}_{p(v_i^-)} \log \sum_{i=1}^M \exp(f(v_i^1)f(v_i^-)) \right] - 3\delta_{aug}^2 - \delta_{aug} - \mathbb{E}_{p(v^0,y)} ||f(v_y^0) - \mu_y||$$

$$\quad - \log M - \frac{1}{2} \sum_{j=1}^m \mathrm{Var}(f_j(v^-|y)) - A(M) + \log K - \delta_{aug} - e \sum_{j=1}^n \mathrm{Var}(\mu_j)$$

$$= \mathcal{L}_{\mathrm{NCE}} - 3\delta_{aug}^2 - 2\delta_{aug} - \log \frac{M}{K} - \frac{1}{2} \sum_{j=1}^m \mathrm{Var}(f_j(v^-|y)) - A(M) - e \sum_{j=1}^n \mathrm{Var}(\mu_j) - \mathbb{E}_{p(v^0,y)} ||f(v_y^0) - \mu_y||$$

$$\overset{(g)}{\geq} \mathcal{L}_{\mathrm{NCE}} - 3\delta_{aug}^2 - 2\delta_{aug} - \log \frac{M}{K} - \frac{1}{2} \mathrm{Var}(f(v^+)|y) - \sqrt{\mathrm{Var}(f(v^0)|y)} - O(M^{-\frac{1}{2}}) - e \mathrm{Var}(\mu_y).$$

(g) This holds because, $v^-$ is randomly selected from $v^+$ and,

$$\sum_{j=1}^{m} \mathrm{Var}(f_j(v^-|y))$$

$$= \sum_{j=1}^{m} \mathbb{E}_{p(y)}\mathbb{E}_{p(v|y)}(f_j(v^+) - \mathbb{E}_{p(v'|y)}f_j(v'))^2$$

$$= \mathbb{E}_{p(y)}\mathbb{E}_{p(v|y)} \sum_{j=1}^{m} (f_j(v^+) - \mathbb{E}_{v'}f_j(v'))$$

$$= \mathbb{E}_{p(y)}\mathbb{E}_{p(v|y)}||f(v) - \mathbb{E}_{v'}f(v')||^2$$

$$= \mathrm{Var}(f(v^+)|y).$$

And similarly, we can get $\sum_{j=1}^{n} \mathrm{Var}(\mu_j) = \mathrm{Var}(\mu_y)$.

## A.3 PROOF OF COROLLARY 3.1

For $\mathrm{Var}(f(v_y^0|y))$, we can use positive pair difference and the intrinsic property of model and data to express.

$$\mathrm{Var}(f(v_y^0|y)) = \mathbb{E}_{p(y)}\mathbb{E}_{p(v_y^0|y)}||f(v_y^0) - \mu_y||^2$$

$$= \mathbb{E}_{p(y)}\mathbb{E}_{p(v_y^0|y)} \left[ (f(v_y^0) - \mu_y)^T(f(v_y^0) - \mu_y)) \right]$$

$$= \mathbb{E}_{p(y)}\mathbb{E}_{p(v_y^0|y)} \left[ f(v_y^0)^T f(v_y^0) + \mu_y^T \mu_y - 2f(v_y^0)^T \mu_y \right]$$

$$\leq \mathbb{E}_{p(y)}\mathbb{E}_{p(v_y^0|y)} \left[ 2 - 2f(v_y^0)^T \mu_y \right]$$

$$\overset{(h)}{\leq} \mathbb{E}_{p(y)}\mathbb{E}_{p(v_y^0|y)} \left[ 2 - 2(1 - \frac{1}{3}\delta_{aug}^2 - \frac{2}{3}\delta_{aug}\delta_{y^+} - \frac{1}{2}\delta_{y^+}^2) \right]$$

$$= \mathbb{E}_{p(y)}\mathbb{E}_{p(v_y^0|y)} \left[ \frac{2}{3}\delta_{aug}^2 + \frac{4}{3}\delta_{aug}\delta_{y^+} + \delta_{y^+}^2) \right]$$

$$\leq \frac{2}{3}\delta_{aug}^2 + \frac{4}{3}\delta_{aug}L\epsilon_0 + L^2\epsilon_0^2,$$

where $\epsilon_0 = \mathbb{E}_{p(y)}\mathbb{E}_{p(v_i^0,v_j^0|y)}||v_i^0 - v_j^0||$ and $L$ is the Lipschitz constant, so $\delta_{y^+}^2 = \mathbb{E}_{p(y,i,j)}||f(v_{y,i}^0) - f(v_{y,j}^0)||^2 \leq (L\epsilon_0)^2$.

Then we can easily get that,

$$\mathrm{Var}(f(v_y^+)|y) = \mathbb{E}_{p(y)}\mathbb{E}_{p(v_y^-|y)}||f(v_y^+) - \mu_y||^2$$

$$\leq \mathbb{E}_{p(y)}\mathbb{E}_{p(v_y^+|y)}(||f(v_y^+) - f(v_y^0)|| + ||f(v_y^0) - \mu_y||)^2$$

$$= \mathbb{E}_{p(y)}\mathbb{E}_{p(v_y^+|y)}||f(v_y^+) - f(v_y^0)||^2 + \mathbb{E}_{p(y)}\mathbb{E}_{p(v_y^+|y)}||f(v_y^0) - \mu_y||)^2$$

$$\quad + 2\mathbb{E}_{p(y)}\mathbb{E}_{p(v_y^+|y)}||f(v_y^+) - f(v_y^0)|| \cdot ||f(v_y^0) - \mu_y||$$

$$\leq \delta_{aug}^2 + \mathrm{Var}(f(v_y^0)|y) + 2\delta_{aug}\sqrt{\mathrm{Var}(f(v_y^0)|y)}$$

$$= (\delta_{aug} + \sqrt{\mathrm{Var}(f(v_y^0)|y)})^2.$$

(h) We use Theorem 2.3.

And $\mathrm{Var}(\mu_y)$ can also be expressed by intrinsic properties.

$$\mathrm{Var}(\mu_y) = \mathbb{E}_{p(y)}||\mu_y - \mu||^2$$

$$= \mathbb{E}_{p(y)}||\mu_y - f(v_y^*) + f(v_y^*) - \mu||^2$$

$$\leq \mathbb{E}_{p(y)}(||\mu_y - f(v_y^*)|| + ||f(v_y^*) - \mu||)^2$$

$$= \mathbb{E}_{p(y)}||\mathbb{E}_{p(v_y|y)}f(v_y) - f(v_y^*)||^2 + \mathbb{E}_{p(y)}||f(v_y^*) - \mathbb{E}_{p(v)}f(v)||^2$$

$$+ 2\mathbb{E}_{p(y)}(||\mathbb{E}_{p(v_y|y)}f(v_y) - f(v_y^*)|| \cdot ||f(v_y^*) - \mathbb{E}_{p(v)}f(v)||)$$
$$= \mathbb{E}_{p(y)}||\mathbb{E}_{p(v_y|y)}[f(v_y) - f(v_y^*)]||^2 + \mathbb{E}_{p(y)}||\mathbb{E}_{p(v)}[f(v_y^*) - f(v)]||^2$$
$$+ 2\mathbb{E}_{p(y)}(||\mathbb{E}_{p(v_y|y)}[f(v_y) - f(v_y^*)]|| \cdot ||\mathbb{E}_{p(v)}[f(v_y^*) - f(v)]||)$$
$$\leq \mathbb{E}_{p(y)}\mathbb{E}_{p(v_y|y)}||f(v_y) - f(v_y^*)||^2 + \mathbb{E}_{p(y)}\mathbb{E}_{p(v)}||f(v_y^*) - f(v)||^2$$
$$+ 2\mathbb{E}_{p(y)}(\mathbb{E}_{p(v_y|y)}||f(v_y) - f(v_y^*)|| \cdot ||f(v_y^*) - f(v)||)$$
$$\leq L^2\epsilon_1^2 + L^2\epsilon_2^2 + 2L^2\epsilon_1\epsilon_2$$
$$= L^2(\epsilon_1 + \epsilon_2)^2,$$

where $v_y^*$ could be any node of class $y$, and $\epsilon_1 = \mathbb{E}_{p(v,y)}||v_y - v_y^*||$, $\epsilon_2 = \mathbb{E}_{p(y)}\mathbb{E}_{p(v)}||v - v_y^*||$.

$$\hat{\mathcal{L}}_{\text{CE}} \geq \mathcal{L}_{\text{NCE}} - 3\delta_{aug}^2 - 2\delta_{aug} - \log\frac{M}{K} - \frac{1}{2}\text{Var}(f(v^-)|y) - \sqrt{\text{Var}(f(v^0)|y)} - O(M^{-\frac{1}{2}}) - e\,\text{Var}(\mu_y)$$

$$\geq \mathcal{L}_{\text{NCE}} - 3\delta_{aug}^2 - 2\delta_{aug} - \log\frac{M}{K} - \frac{1}{2}(\delta_{aug} + \sqrt{\text{Var}(f(v_y^0)|y)})^2 - \sqrt{\text{Var}(f(v_y^0)|y)} - O(M^{-\frac{1}{2}}) - eL^2(\epsilon_1 + \epsilon_2)^2$$

$$= \mathcal{L}_{\text{NCE}} - 3\delta_{aug}^2 - 2\delta_{aug} - log\frac{M}{K} - \frac{1}{2}\delta_{aug}^2 - (\delta_{aug} + 1)\sqrt{\text{Var}(f(v_y^0)|y)}$$

$$- \frac{1}{2}\text{Var}(f(v_y^0)|y)) - O(M^{-\frac{1}{2}}) - eL^2(\epsilon_1 + \epsilon_2)^2$$

$$= \mathcal{L}_{\text{NCE}} - g(\delta_{aug}) - \log\frac{M}{K} - O(M^{-\frac{1}{2}}),$$

where $g(\delta_{aug}) = \frac{23}{6}\delta_{aug}^2 + \frac{1}{2}L^2\epsilon_0^2 + eL^2(\epsilon_1 + \epsilon_2)^2 + 2\delta_{aug} + \frac{2}{3}\delta_{aug}L\epsilon_0 + (\delta_{aug} + 1)\sqrt{\frac{2}{3}\delta_{aug}^2 + \frac{4}{3}\delta_{aug}L\epsilon_0 + L^2\epsilon_0^2}$.

According to Oord et al. (2018), we get,
$$I(f(v_i^1), f(v_i^2)) \geq \log(M) - \mathcal{L}_{\text{NCE}},$$
$$\mathcal{L}_{\text{NCE}} \geq \log(M) - I(f(v_i^1), f(v_i^2)).$$

Therefore, we can reformulate Theorem 2.5 as below:
$$\hat{\mathcal{L}}_{\text{CE}} \geq \log(M) - I(f(v_i^1), f(v_i^2)) - g(\delta_{aug}) - \log\frac{M}{K} - O(M^{-\frac{1}{2}})$$
$$= \log(K) - I(f(v_i^1), f(v_i^2)) - g(\delta_{aug}) - O(M^{-\frac{1}{2}}).$$

## A.4 PROOF OF COROLLARY 3.3

Corallary 3.3 could be simply proved below:
$$\mathbb{E}_{p(v_i^1, v_i^2)}||f(v_i^1) - f(v_i^2)||^2 = \mathbb{E}_{p(v_i^1, v_i^2)}[(f(v_i^1)^T - f(v_i^2)^T)(f(v_i^1) - f(v_i^2))]$$
$$= \mathbb{E}_{p(v_i^1, v_i^2)}[2 - 2f(v_i^1)^T f(v_i^2)]$$
$$= 2 - \frac{2}{N}tr((H^1)^T H^2)$$
$$\stackrel{(1)}{=} 2 - \frac{2}{N}\sum_i \theta_i \lambda_i' \lambda_i''.$$

So $(\mathbb{E}_{p(v_i^1, v_i^2)}||f(v_i^1) - f(v_i^2)||)^2 \leq \mathbb{E}_{p(v_i^1, v_i^2)}||f(v_i^1) - f(v_i^2)||^2 = 2 - \frac{2}{N}\sum_i \theta_i \lambda_i' \lambda_i''$, then $\mathbb{E}_{p(v_i^1, v_i^2)}||f(v_i^1) - f(v_i^2)|| \leq \sqrt{2 - \frac{2}{N}\sum_i \theta_i \lambda_i' \lambda_i''}$.

(1) is suggested by Liu et al. (2022), $tr((H^1)^T H^2)$ could be represented as $\sum_i \theta_i \lambda_i' \lambda_i''$.

As we know that,
$$\mathbb{E}_{p(v_i^1, v_i^2)}||f(v_i^1) - f(v_i^2)|| \leq \mathbb{E}_{p(v_i^1, v_i^2)}(||f(v_i^1) - f(v_i^0)|| + ||f(v_i^0) - f(v_i^2)||) \leq 2\delta_{aug}.$$

Then, we can get:
$$2\delta_{aug} \geq \mathbb{E}_{p(v_i^1, v_i^2)}||f(v_i^1) - f(v_i^2)|| \geq \sqrt{2 - \frac{2}{N}\sum_i \theta_i \lambda_i' \lambda_i''}. \tag{8}$$

## A.5 PROOF OF LEMMA B.1

From Stewart (1990), we know the following equation:

$$\Delta \lambda_i = \lambda_i^{'} - \lambda_i = u_i^T \Delta A u_i - \lambda_i u_i^T \Delta D u_i + O(||\Delta A||).$$

So we can calculate the difference between $\lambda_i^{'}, \lambda_i^{''}$ and $\lambda_i$,

$$\Delta \lambda_i = \sum_m (\sum_n u_i[n] \Delta A[m][n]) u_i[m] - \lambda_i \sum_m u_i[m] \Delta D[m] u_i[m] + O(||\Delta A||)$$
$$= \sum_{m,n} u_i[m] u_i[n] \Delta A[m][n] - \lambda_i \sum_{m,n} u_i[m] \Delta A[m][n] u_i[m] + O(||\Delta A||).$$

And we can directly calculate $\lambda_i^{'} - \lambda_i^{''}$ as below:

$$\lambda_i^{'} - \lambda_i^{''} = \Delta \lambda_i^{'} - \Delta \lambda_i^{''}$$
$$= \sum_{m,n} u_i[m] u_i[n] \Delta \widehat{A}[m][n] - \lambda_i \sum_{m,n} u_i[m] \Delta \widehat{A}[m][n] u_i[m]$$
$$= \sum_{m,n} u_i[m] \Delta \widehat{A}[m][n] (u_i[n] - \lambda_i u_i[m]).$$

# B GCL METHODS WITH SPATIAL AND SPECTRAL AUGMENTATION

## B.1 SPATIAL AUGMENTATION

Most augmentation methods are applied to explicitly or implicitly increase mutual information while maintain high positive pair difference. GRACE simply adjusts this by changing the drop rate of features and edges. AD-GCL (Suresh et al., 2021) directly uses the optimization objective $min_{\{aug\}} max_{\{f \in F\}} I(f(v), f(aug(v)))$ to search for a stronger augmentation.

And GCA (Zhu et al., 2021) could always perform better than random drop. This is mainly because GCA calculates node importance and masks those unimportant to increase mutual information. Also they use $p_\tau$ as a cut-off probability, so for those unimportant features/edges, all of them share the same drop probability $p_\tau$. By setting a large $p_\tau$, GCA can reduce the drop probability for the least important features/edges and drop more relatively important ones to achieve a trade-off between mutual information and positive pair difference.

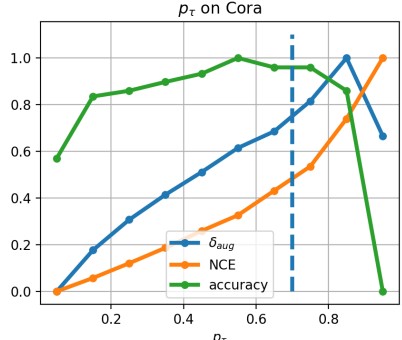

Figure 4: influence of $p_\tau$ on Cora (all the data are normalized for better visualization)

From Figure 4, we could clearly see that, as $p_\tau$ increases, positive pair difference and $\mathcal{L}_{NCE}$ are increasing, and leads to a better downstream performance, than when $p_\tau$ becomes too large, we got a trivial solution. And in the code provided by the author, $p_\tau$ is set to 0.7. So GCA performances well on downstream tasks not only because its adaptive augmentation, but also its modification on positive pair difference.

## B.2 SPECTRAL AUGMENTATION

Furthermore, we can demonstrate that lots of spectral augmentations follow this schema to improve downstream performance. Liu et al. (2022) proposes that increasing the number of high-frequency drops leads to better performance. This is because high-frequency parts are associated with higher coefficients $\lambda_i$, so increasing the number of high-frequency drops can have a stronger increasement on $\delta_{aug}$, resulting in better performance.

**Lemma B.1** (Change of Spectrum). *if we assume that $A' = A + \Delta A_1$, $A'' = A + \Delta A_2$, $\lambda_i'$, $\lambda_i''$ is the $i^{th}$ eigenvalue of $A'$ and $A''$, respectively. $\Delta \hat{A} = A' - A''$, and $u_i$ is the corresponding eigenvector.*

$$\lambda_i' - \lambda_i'' = \sum_{m,n} u_i\,[m]\,\Delta \widehat{A}\,[m]\,[n]\,(u_i\,[n] - \lambda_i u_i\,[m]).$$

Lemma B.1 is proved in Appendix A.5. Lin et al. (2022) propose to maximize the spectral difference between two views, but Lemma B.1 shows that difference between spectrum is highly correlated with the original magnitude, so it is actually encouraging more difference in large $|\lambda_i|$. But rather than just drop information, they try to improve the spectrum of first view, and decrease the other view. if we simply assume $\lambda_i' = \lambda_i + n$, $\lambda_i'' = \lambda_i - n$, then $\lambda_i'\lambda_i'' = \lambda_i^2 - n^2 \le \lambda_i^2$, so this could also be explained by positive pair difference increasement.

## C  EXPERIMENTS

### C.1  DATASETS AND EXPERIMENTAL DETAILS

We choose the six commonly used Cora, CiteSeer, PubMed, DBLP, Amazon-Photo and Amazon-Computer for evaluation. The first four datasets are citation networks (Sen et al., 2008; Yang et al., 2016; Bojchevski & Günnemann, 2017), where nodes represent papers, edges are the citation relationship between papers, node features are comprised of bag-of-words vector of the papers and labels represent the fields of papers. In Amazon-Photos and Amazon-Computers (Shchur et al., 2018), nodes represent the products and edges means that the two products are always bought together, the node features are also comprised of bag-of words vector of comments, labels represent the category of the product.

We use 2 layers of GCNConv as the backbone of encoder, we use feature/edge drop as data augmentation, the augmentation is repeated randomly every epoch, and InfoNCE loss is conducted and optimized by Adam. After performing contrastive learning, we use logistic regression for downstream classification the solver is liblinear, and in all 6 datasets we randomly choose 10% of nodes for training and the rest for testing.

Table 2: Dataset statistics

| Dataset | Nodes | Edges | Features | Classes |
|---|---|---|---|---|
| Cora | 2,708 | 5,429 | 1,433 | 7 |
| Citeseer | 3,327 | 4,732 | 3,703 | 6 |
| Pubmed | 19,717 | 44,338 | 500 | 3 |
| DBLP | 17,716 | 105,734 | 1,639 | 4 |
| Amazon-Photo | 7,650 | 119,081 | 745 | 8 |
| Amazon-Computers | 13,752 | 245,861 | 767 | 10 |

Table 3: Dataset download links

| Dataset | Download Link |
|---|---|
| Cora | https://github.com/kimiyoung/planetoid/raw/master/data |
| Citeseer | https://github.com//kimiyoung/planetoid/raw/master/data |
| Pubmed | https://github.com/kimiyoung/planetoid/raw/master/data |
| DBLP | https://github.com/abojchevski/graph2gauss/raw/master/data/dblp.npz |
| Amazon-Photo | https://github.com/shchur/gnn-benchmark/raw/master/data/npz/amazon_electronics_photo.npz |
| Amazon-Computers | https://github.com/shchur/gnn-benchmark/raw/master/data/npz/amazon_electronics_computers.npz |

And the publicly available implementations of Baselines can be found at the following URLs:

- GCN: https://github.com/tkipf/gcn

- GAT: https://github.com/PetarV-/GAT

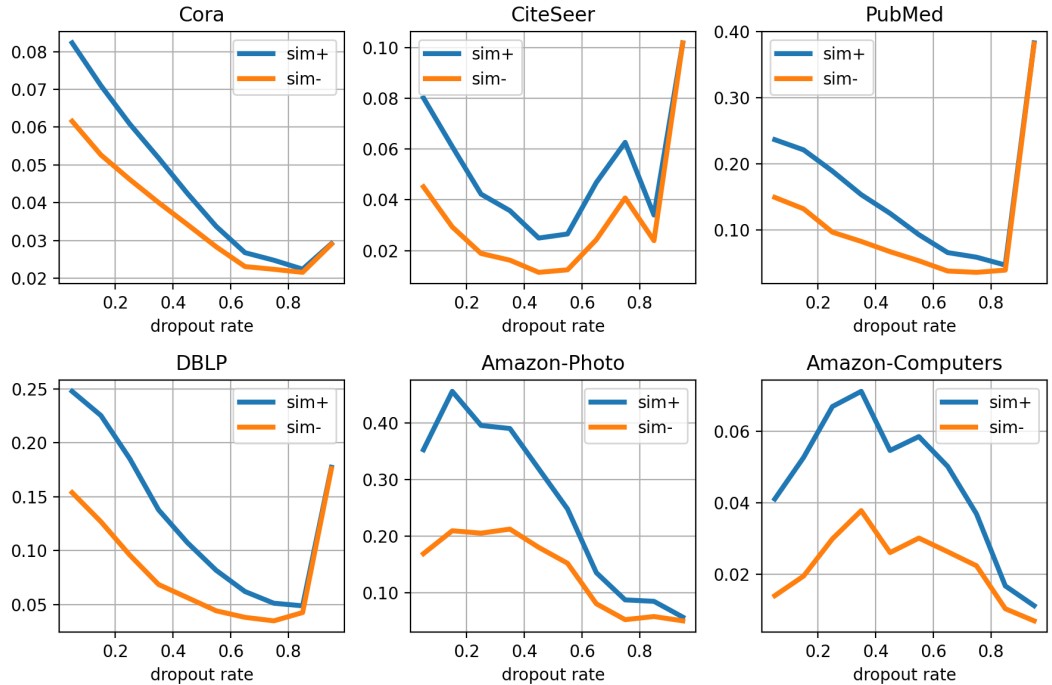

Figure 5: sim+ represents the positive pair similarity $f(v_i^1)^T f(v_i^2)$, and sim− is negative pair similarity $f(v_i^1)^T f(v_i^-)$, the x-axis stands for dropout rate on edges

- GRACE: https://github.com/CRIPAC-DIG/GRACE
- GCA: https://github.com/CRIPAC-DIG/GCA
- AD-GCL: https://github.com/susheels/adgcl
- GCS: https://github.com/weicy15/GCS
- SpCo: https://github.com/liun-online/SpCo

## C.2 CHANGE ON POSITIVE/NEGATIVE PAIR SIMILARITY

The InfoNCE loss $\mathcal{L}_{\text{NCE}}$ can be written as $\mathcal{L}_{\text{NCE}} = \mathbb{E}_{p(v_i^1,v_i^2)}\mathbb{E}_{p(v_i^-)}\left[-\log\frac{\exp(f(v_i^1)^T f(v_i^2))}{\sum_{\{v_i^-\}}\exp(f(v_i^1)^T f(v_i^-))}\right]$, and when we perform stronger augmentation, $f(v_i^1)^T f(v_i^2)$ would be hard to maximize, and the model will try to minimize $f(v_i^1)^T f(v_i^-)$ harder. From Figure 5, when the augmentation gets stronger, negative and positive pair similarity both decreases, so the class separating performance is enhanced.

## C.3 HYPERPARAMETER SETTING

Table 4: Hyperparameters settings

| Dataset | Learning rate | Weight decay | num layers | $\tau$ | Epochs | Hidden dim | Activation |
|---|---|---|---|---|---|---|---|
| Cora | $5^{-4}$ | $10^{-6}$ | 2 | 0.4 | 200 | 128 | ReLU |
| Citeseer | $10^{-4}$ | $10^{-6}$ | 2 | 0.9 | 200 | 256 | PReLU |
| Pubmed | $10^{-4}$ | $10^{-6}$ | 2 | 0.7 | 200 | 256 | ReLU |
| DBLP | $10^{-4}$ | $10^{-6}$ | 2 | 0.7 | 200 | 256 | ReLU |
| Amazon-Photo | $10^{-4}$ | $10^{-6}$ | 2 | 0.3 | 200 | 256 | ReLU |
| Amazon-Computers | $10^{-4}$ | $10^{-6}$ | 2 | 0.2 | 200 | 128 | RReLU |

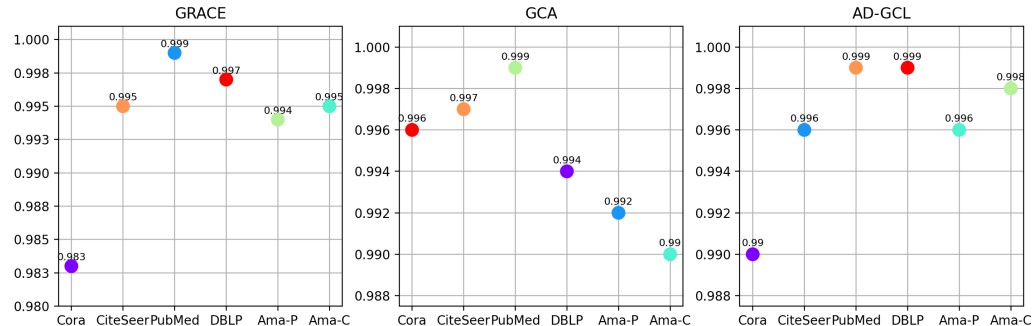

Figure 6: Percentage of positive $\theta$

The hyperparameter settings is shown in Table 4, other hyperparameter correlated to only one algorithm are set the same as the original author. The hyperparameter in our methods retain rate $\xi$ and spectrum change magnitude $\alpha$, we select them from 0.05 to 0.45 and from -0.1 to 0.01, respectively.

### C.4 VALUE OF $\theta$S

As defined by Liu et al. (2022), $\theta$s are actually linear combination of the eigenvalues of adjacency matrix $A$. To demonstrate what $\theta$s actually are, we first focus on the assumption below.

**Assumption C.1** (High-order Proximity). $M = w_0 + w_1 A + w_2 A^2 + \cdots + w_q A^q$, where $M = X^1 W \cdot W^T (X^2)^T$, $A^i$ means matrix multiplications between $i$ $A$s, and $w_i$ is the weight of that term.

Where $X^1, X^2$ indicates the feature matrix of graph $\mathcal{G}^1, \mathcal{G}^2$, $W$ stands for the parameter of the model, so $M = X^1 W \cdot W^T (X^2)^T$ means embedding similarity between two views, and could be roughly represented by the weighted sum of different orders of $A$. Furthermore, we have that:

$$
\begin{cases}
w_0 + w_1 \lambda_1 + \cdots + w_q \lambda_1^q = \theta_1 \\
w_0 + w_1 \lambda_2 + \cdots + w_q \lambda_2^q = \theta_2 \\
\qquad \cdots \\
w_0 + w_1 \lambda_N + \cdots + w_q \lambda_N^q = \theta_N,
\end{cases}
$$

where $\lambda_1, ..., \lambda_N$ is N eigenvalues of the adjacency matrix $A$.

So we know that $\theta$s are actually linear combination of $\lambda$s. As the model is trained to minimize $\mathcal{L}_{\text{NCE}}$, $\theta$s are expected to increase, and we can simply set $w_0, w_2, ..., w_{2(q/2)}$ to be positive and other $w_i$ to 0, then we can get $\theta$s that are all positive, and the model would easily find better $w$s.

Therefore, we can say that in the training process, $\theta$s are mostly positive, and the experiments shown in Figure 7 indicate it to be true.

### C.5 CHANGES ON THE SPECTRUM

From Figure 7(a), we can see that, when the algorithm is training, $\theta$s are mostly increasing gradually, and when we perform spectrum augmentation, $\theta$s will not increase as before, increasing number of $\theta$ is close even smaller to decreasing ones. Then we take a step back on those decreasing ones, result in increasing $\theta$s again in the next epoch. Therefore, what we do is actually perform augmentation to maximize positive pair difference first, then maximize the mutual information after spectrum augmentation. The idea is actually similar AD-GCL, but we use $\theta$s to indicate whether the augmentation is too much, so we get a more reasonable result. Figure 7(b) and (c) shows that as the training goes, the change on larger magnitude eigenvalues are also more significant, causing the spectrum to be smoother.

Also there is one thing to notice that when we perform spectrum smoothen method, we are indirectly changing the edge weights, causing the augmentation being weaker or stronger as drop an edge with

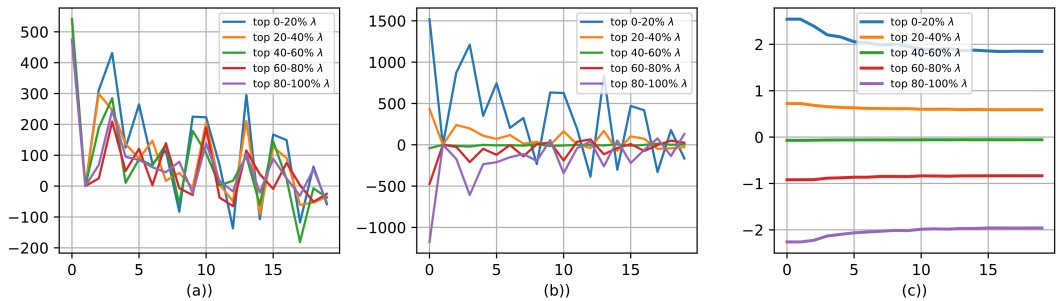

Figure 7: As we perform spectrum augmentation each 10 epochs, the x-axis is epoch/10, the y-axis of the left figure is number of decreasing $\lambda$s minus number of increasing $\lambda$s; for the middle one, y-axis stands for how much $\lambda$s averagely decreases; and the right one is the average value of $\lambda$.

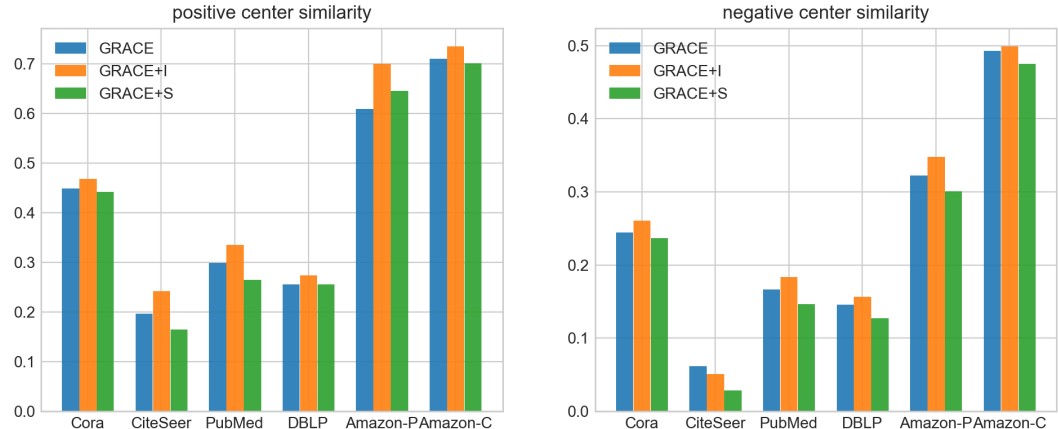

Figure 8: similarity of nodes between its positive center and negative center, GRACE stands for the pure GRACE, GRACEI stands for GRACE with information augmentation, and GRACES stands for GRACE with spectrum augmentation

weight of 1 is different than drop an edge with weight $1 + noise$. To reduce its influence, we conduct extra augmentation or recovery based on the average weight change.

### C.6 CENTER SIMILARITY

As we mentioned earlier, GCL mainly contributes to downstream tasks by decreasing the negative center similarity while maintaining a relatively high similarity to the positive center. We propose two methods: one that increases mutual information between two views while keeping a high positive pair difference by masking more unimportant edges or features. This allows the model to learn more useful information, which forces nodes close to its positive center. The other method tries to increase positive pair difference while maintaining a relatively high mutual information, so it may not learn as much useful information. However, by increasing the positive pair difference, it forces the model to separate nodes from different classes further apart. In summary, the first method brings nodes of the same class closer together, while the second method separates nodes from different classes further apart just as shown in Figure 8.

### C.7 HYPERPARAMETER SENSITIVITY

**Analysis of retain rate**. Retain rate controls how many important features/edges we kept, and how many unimportant ones dropped. We can see from Figure 9 that AD-GCL benefits from a larger retain rate as it is designed to minimize the mutual information, and lots of vital structures

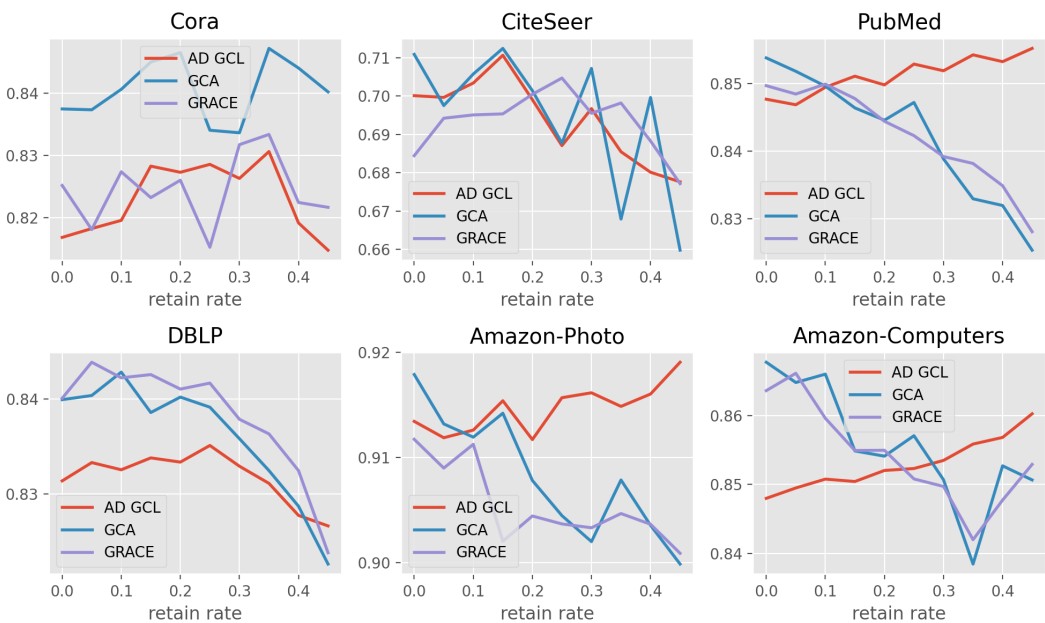

Figure 9: accuracy on downstream tasks with different $\alpha$

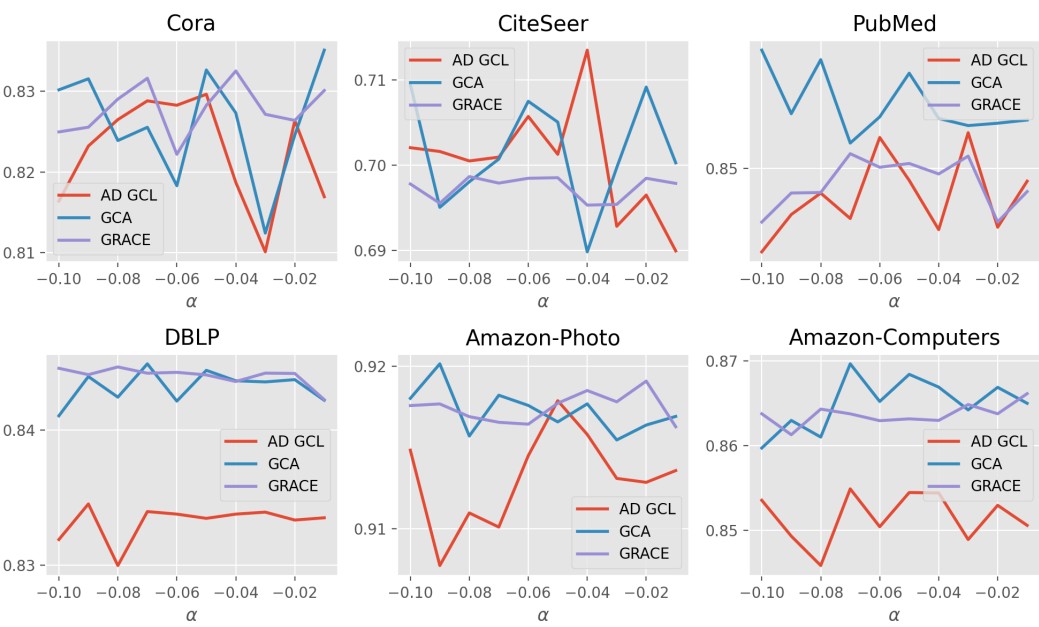

Figure 10: accuracy on downstream tasks with different retain rate

are dropped. And large datasets like PubMed, DBLP benefits less, it is mainly because a graph with more edges are more likely to maintain enough information than graph with little edges. For example, after a 30% dropout on edges, a graph with 1000 edges would still kept enough information for downstream tasks, but a graph with 10 edges would probably lose some vital information.

**Analysis of** $\alpha$. $\alpha$ controls how much $|\lambda|$ will decrease, as we take a step back when the $|\lambda|$ decreases too much, the hyperparameter $\alpha$ does not matter so much. But as shown in Figure 10, it still performs more steady on large graphs as a wrong modification on a single $\lambda$ matters less than on small graphs.

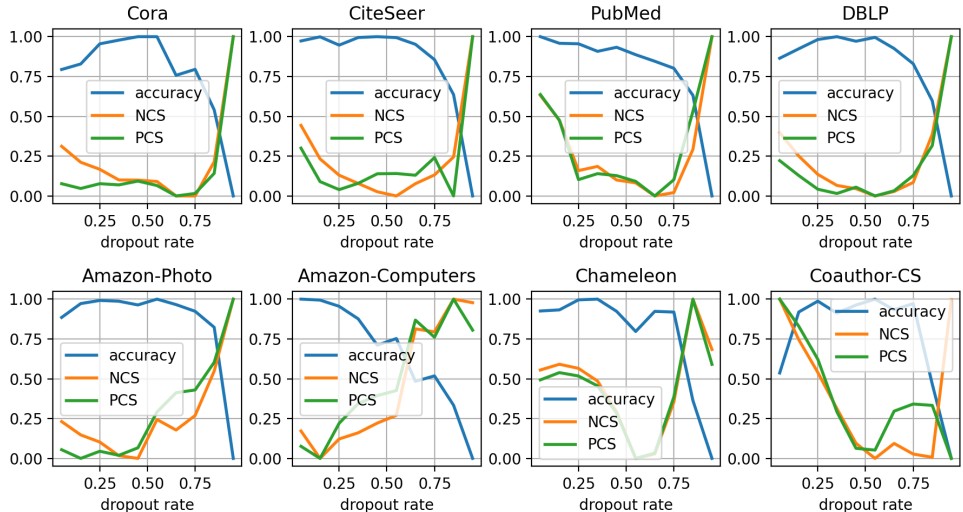

Figure 11: More experiments on PCS and NCS, the settings follow Figure 1, but the detailed data is slightly different because of randomness, but it shows similar tendency

## C.8 PCS, NCS AND DOWNSTREAM PERFORMANCE

More experiments are conducted on various of datasets to show that our finding could be generalized rather that limited to few datasets. It is worth noticing that we conduct two identical experiments, and they are both average of 10 runs, so Figure 11 may slightly differ from Figure 1, but they show similar tendency that with the dropout rate increasing, the downstream accuracy increases first and decreases when

Table 5: results on image datasets

| dataset | item | 0.2 | 0.4 | 0.6 | 0.8 | 1.0 |
|---------|------|-----|-----|-----|-----|-----|
| stl10 | PCS | 0.48 | 0.47 | 0.43 | 0.47 | 0.47 |
| | NCS | 0.11 | 0.09 | 0.05 | 0.04 | 0.04 |
| | Acc | 0.68 | 0.69 | 0.71 | 0.73 | 0.73 |
| cifar10 | PCS | 0.4 | 0.39 | 0.37 | 0.41 | 0.42 |
| | NCS | 0.029 | 0.031 | 0.027 | 0.025 | 0.021 |
| | Acc | 0.71 | 0.71 | 0.75 | 0.77 | 0.77 |

the augmentation is too strong. The accuracy of some datasets like CiteSeer, PubMed and Amazon-Computers seems to keep decreasing, this is mainly because we start the dropout rate with 0.05, and when the dropout rate changes from 0.00 to 0.05 in Figure 12, we can also observe increasing tendency. And those experiments show that when the downstream accuracy increases, the positive center similarity are sometimes decreasing, and the better downstream performance is mainly caused by the decreasing similarity of negative center.

We also conduct experiments on images to verify our theory, we control the magnitude of augmentation by adjusting the color distortion strength, and the results are normalized by Min-Max normalization. From Figure 13, we can observe that the downstream performance is also closely correlated with negative center similarity especially when the color distortion strength changes from 0.2 to 0.6 the positive center similarity decreases while downstream performance is increasing, but when color distortion is greater

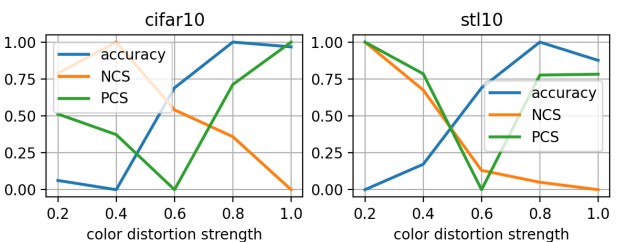

Figure 13: relationship of PCS, NCS and downstream performance on images, the data is normalized

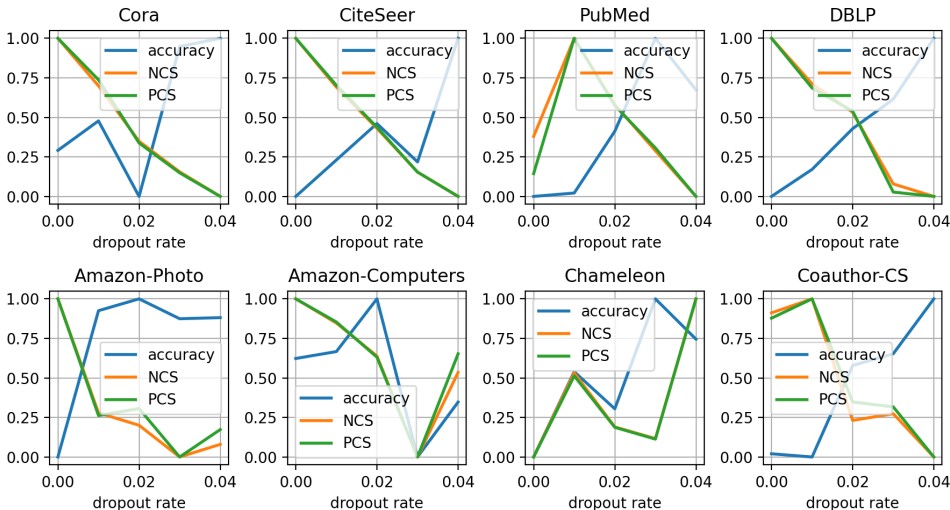

Figure 12: More experiments on PCS and NCS, but the drop rate is set to 0.01-0.05.

than 0.6 the positive center similarity also tends to increase. This aligns with our finding in Theorem 2.3 that with the augmentation gets stronger the negative center similarity is decreasing while the positive center similarity does not change in specific pattern. Also the color distortion is not strong enough to change the label information, so the downstream performance keeps increasing with stronger augmentation.

## C.9 CHANGE OF $\delta_{aug}$ AND LABEL CONSISTENCY

To verify how is $\delta_{aug}$ changing with stronger augmentation, we use drop rate of edges/features as data augmentation, and find that when the drop rate increases, $\delta_{aug}$ also tends to increase. Also to verify the view invariance assumption, we first train a well conditioned model and use its prediction as $p(v_i)$, then we change the drop rate and calculate new $p'(v_i)$, then we can observe from Figure 14 that though the KL divergence is increasing

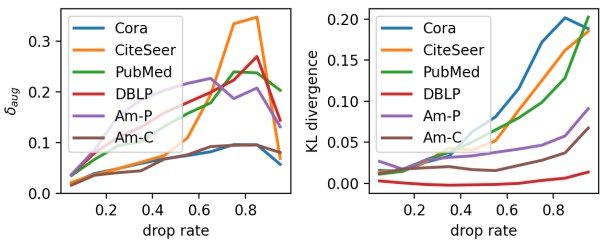

Figure 14: relationship between $\delta_{aug}$, KL divergence and augmentation

with drop rate, it remains quite small magnitude, so the label is consistent with data augmentation.

## C.10 TIME COMPLEXITY AND ERROR BAR

Table 6: The time consumption (seconds) of algorithms

|         | Cora  | CiteSeer | PubMed | DBLP  | Amazon-P | Amazon-C |
|---------|-------|----------|--------|-------|----------|----------|
| GRACE   | 8.02  | 10.08    | 62.37  | 56.89 | 19.05    | 28.71    |
| GRACE+I | 10.74 | 13.49    | 68.97  | 62.8  | 22.67    | 29.61    |
| GRACE+S | 9.61  | 12.46    | 78.11  | 69.44 | 21.13    | 36.94    |

From Table 6, we can observe that the information augmentation method achieve better performance with only few more time consuming, this is mainly because we do not calculate the importance of features/edges every epoch like GCS (Wei et al., 2023), we only calculate it once and use the same

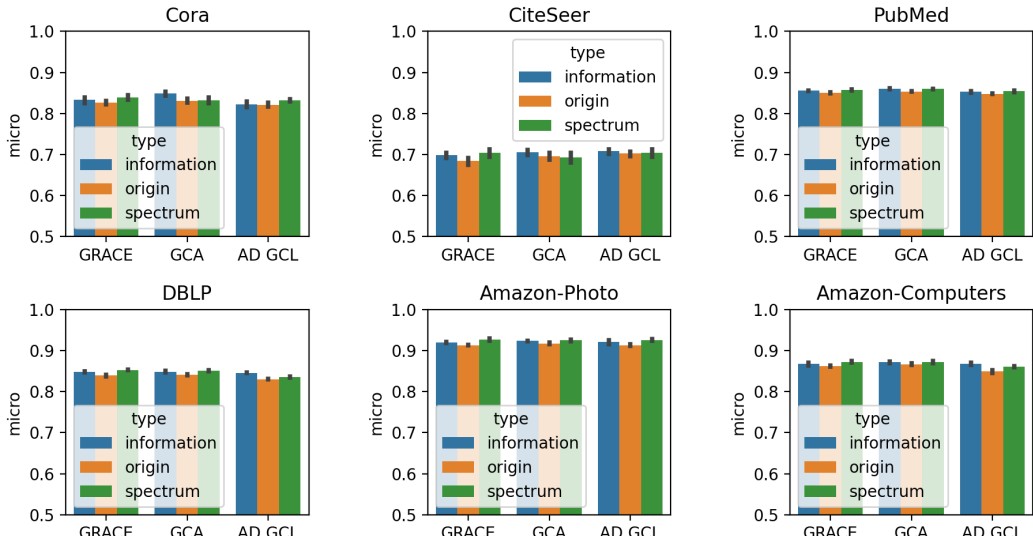

Figure 15: The error bar of algorithms

importance for the following training. However, the spectrum augmentation method consumes more time on large graphs like PubMed and DBLP, this is mainly we explicitly change the spectrum and calculate the new adjacency matrix, which could be replaced by some approximation methods but to prevent interference from random noise and show that Theorem 3.2 is meaningful, we still conduct eigen decomposition, but it is worth mentioning that the time complexity could be reduced by some approximation methods (Liu et al., 2022).

The error bar is reported in Figure 15, the experiments are conducted repeatedly for 10 times, we can observe that both the information augmentation and spectrum augmentation achieve better results, and they performs stably.

## D   RELATED WORK

**Graph Contrastive Learning.** Graph Contrastive Learning has shown its superiority and lots of researcher are working on it. DGI (Veličković et al., 2018) contrasts between local node embeddnig and the global summary vector; GRACE (Zhu et al., 2020), GCA (Zhu et al., 2021) and GraphCL (You et al., 2020) randomly drop edges and features; AD-GCL (Suresh et al., 2021) and InfoGCL Xu et al. (2021) learn an adaptive augmentation with the help of different principles. In theoretical perspective, Liu et al. (2022) correlates the InfoNCE loss with graph spectrum, and propose that augmentation should be more focused on high frequency parts. Guo et al. (2023) further discuss that contrastive learning in graph is different with images. Lin et al. (2022) thinks that augmentation maximize the spectrum difference would help, and Yuan et al. (2022) analyse GCL with information theory.

**Contrastive Learning Theory.** By linking downstream classification and contrastive learning objectives, Arora et al. (2019) propose a theoretical generalization guarantee. Ash et al. (2021) further explore how does the number of negative samples influence the generalization. And Tian et al. (2020); Wang et al. (2022a) further discuss what kind of augmentation is better for downstream performance. Then Wang & Isola (2020) propose that perfect alignment and uniformity is the key to success while Wang et al. (2022b) argues augmentation overlap with alignment helps gathering intra-class nodes by stronger augmentation. However, Saunshi et al. (2022) show that augmentation overlap is actually quite rare while the downstream performance is satisfying. So the reason why contrastive learning helps remains a mystery, in this paper we propose that the stronger augmentation mainly helps contrastive learning by separating inter-class nodes, and different from previous works (Wang et al., 2022b; Wang & Isola, 2020; Huang et al., 2021), we do not treat perfect alignment as key to success, instead a stronger augmentation that draw imperfect alignment could help.

