# OpenReview forum: "Perfect Alignment May be Poisonous to Graph Contrastive Learning"
_ICLR.cc/2024/Conference — Submitted to ICLR 2024_

### Official Review · Reviewer_vy1b · 2023-10-30

**Soundness:** 2 fair
**Presentation:** 3 good
**Contribution:** 2 fair
**Rating:** 3
**Confidence:** 5

**Summary:**

This paper discusses the effect of the augmentation on graph contrastive learning (GCL).  Then, this paper proposes that GCL contributes to downstream tasks mainly by separating different classes rather than gathering nodes of the same class. Also, this paper provides both theoretical and empirical evidence to verify the proposed conclusions.

**Strengths:**

1. This paper gives some theoretical analysis.
2. Empirical evidence suggests that GCL contributes to downstream tasks mainly by separating different classes rather than gathering nodes of the same class.

**Weaknesses:**

1. In Theorem 2.3, Equations (2) and (3) are a form of infimum. Then, authors say, “However, the right hand side of Inequality (3) should decrease gradually as both ${\delta _{aug}}$ and ${\delta _{{y^ - }}}$ increase, so the negative center similarity would be more likely to be lower.“ This is problematic because a smaller lower bound does not tell us whether the left side of equation 3 is smaller.
2. The authors say, "Initially, as the dropout rate increases, positive center similarity may decrease sometimes, but downstream performance could be enhanced as negative center similarity decreases much faster." This is problematic. From Figure 1, we can observe that, except for the second subgraph, the performance of the model is basically positively correlated with positive center similarity. The result of the second subgraph may be due to the randomness of the dataset selection, and I suggest that the authors give the average of the results of many different experiments.
3. In Theorem 2.5, the related equation is a form of infimum. Then, the authors say, "So better generalization correlates with worse positive center similarity." This is problematic because a smaller lower bound does not tell us whether the left side of the equation is smaller.
4. From Theorem 2.5, I think we cannot determine why, as the augmentation becomes stronger, negative center similarity decreases while positive center similarity remains unpredictable.  First, when minimizing the NCE loss, the numerator is also changing. So, It's not reasonable to substitute $1 - {\delta _{aug}}$ directly directly for $f{(v_i^1)^T}f(v_i^2)$. Second, even if the right-hand side of the related equation is equal to 0, that doesn't mean the left-hand side of the related equation is equal to 0. Because the right side of the related equation is the lower bound of the left.
5. In Corollary 3.1, the related equation is also a form of infimum. So, even if the right-hand side of equation 4 is equal to 0, that doesn't mean the left-hand side of equation 4 is equal to 0. Because the right side of equation 4 is the lower bound of the left.
6. The improvement in performance is negligible.
7. My personal understanding of the two methods proposed in this paper is as follows: (1) From the perspective of information theory, the traditional GCL is based on learning, so that the mutual information between positive samples is as large as possible. This paper proposes that the mutual information between positive samples can be increased directly by data augmentation. (2) From the perspective of spectrum, the traditional GCL smoothen the spectrum through regularization, while this paper proposes that the mutual information between positive samples can be increased directly through data augmentation. Therefore, the motivation presented in this paper is not convincing.

**Questions:**

Please refer to Weaknesses.

**Details Of Ethics Concerns:**

No ethics concerns

---

> ### Author Response · Authors · 2023-11-19
> **Response to Reviewer vy1b Part 1**
>
> Thank you for taking the time to read our work and insightful comments and suggestions! We are happy to know that you think the idea of GCL contributes to downstream tasks mainly by inter-class separating is interesting. Please find below our comments on the raised issues and questions:
>
> __About the lower bound__:
>
> The lower bound does not enforce a better performane, it shows a potential best solution and when the lower bound becomes smaller, the best solution is better so the model potentially performs better. The upper bound instead shows the worst case, smaller the upper bound means that the model could perform better at the worst case, but the modern model is powerful enough to get away from the worst case and be close the the best case, so the lower bound matters. And we also design algorithms and experiments aims to smaller the lower bound, the results show that it indeed helps, which means that our lower bound is meaningful, and smaller the lower bound does help.
>
>
> >__W1__: In Theorem 2.3, Equations (2) and (3) are a form of infimum. Then, authors say, “However, the right hand side of Inequality (3) should decrease gradually as both $\delta_{aug}$ and $\delta_{y^-}$ increase, so the negative center similarity would be more likely to be lower.“ This is problematic because a smaller lower bound does not tell us whether the left side of equation 3 is smaller.
>
> __Answer__: Indeed, the smaller lower bound does not explicitly tells a smaller left side of equation 3, but it does show that the left side could potentially be lower, so we use the word "more likely", and __we did not make the conclusion only by the theorem__, we perform experiments and those experiments show that negative center similarity is actually decreasing when the augmentation gets stronger. So __we find the negative center similarity might be lower through the theorem, and then perform experiments to show it is actually happening.__
>
> >__W2__: The authors say, "Initially, as the dropout rate increases, positive center similarity may decrease sometimes, but downstream performance could be enhanced as negative center similarity decreases much faster." This is problematic. From Figure 1, we can observe that, except for the second subgraph, the performance of the model is basically positively correlated with positive center similarity. The result of the second subgraph may be due to the randomness of the dataset selection, and I suggest that the authors give the average of the results of many different experiments.
>
> __Answer__: We think there exists some misunderstanding. __Experiments shown in Figure 1 supports our finding that downstream performs correlates with negative center similarity better.__ When we first increase the magnitude of augmentation, the downstream performance tends to increase or stay steady, but the positive center similarity is mostly decreasing, so it shows no positive correlation.
>
> In the first subgraph, positive center similarity stays steady with increasing downstream performance it is the decreasing negative center similarity helps the downstream performance. In the second and fourth subgraph, positive center similarity is decreasing, but the downstream performance steadys steady or even increases as the negative center similarity decreases much faster. In the third subgraph the positive center similarity is decreasing even faster than the negative center similarity, which results in the decreasement in downstream performance. __In both four datasets the downstream performance show great negative correlation with negative center similarity and no evident positive correlation with positive center similarity.__ All the results are average of 10 repeated experiments and we have conducted more experiments on other datasets in Appendix C.8, they also show similar tendency.

---

> ### Author Response · Authors · 2023-11-19
> **Response to Reviewer vy1b Part 2**
>
> >__W3__: In Theorem 2.5, the related equation is a form of infimum. Then, the authors say, "So better generalization correlates with worse positive center similarity." This is problematic because a smaller lower bound does not tell us whether the left side of the equation is smaller.
>
> __Answer__: The lower bound does not explicitly tells whether the left side of the equation is smaller, but it means __the model could potentially achieve better performance__, if the lower bound is large, then the model could never achieve good downstream performance, and our theorem tells that with stronger augmentation, the positive center similarity may be decreased, but the generalization might be better, so downstream performance could be increased. __Also emperical results supports our finding, it means the lower bound is not too loose to be invalid, and the lower bound is meaningful because we derive the bound mathematically, the left side is correlated with the right sise.__
>
> >__W4__: From Theorem 2.5, I think we cannot determine why, as the augmentation becomes stronger, negative center similarity decreases while positive center similarity remains unpredictable. First, when minimizing the NCE loss, the numerator is also changing. So, It's not reasonable to substitute $1-\delta_{aug}$ directly directly for $f(v_i^1)^Tf(v_i^2)$. Second, even if the right-hand side of the related equation is equal to 0, that doesn't mean the left-hand side of the related equation is equal to 0. Because the right side of the related equation is the lower bound of the left.
>
> __Answer__: We think there exists some misunderstanding, **we did not substitute $1-\delta_{aug}$ directly for $f(v_i^1)^Tf(v_i^2)$**, we just want to tell that the numerator $f(v_i^1)^Tf(v_i^2)$ is highly related with the strength of augmentation, a stronger augmentation would make the numerator harder to maximize, and we do not need to focus on how is numerator changing during training, we only want to know the final result (we study how is downstream performance changing with augmentation rather than training). Also we define $\delta_{aug}$ as $\mathbb{E} ||f(v)-f(v^+)||^2$, so $\delta_{aug}$ is actually model related and is also chaning with the model. And the smaller lower bound does not explicitly indicate a better performance, however, as we metioned before, the bound is not that loose and it is meaningful, emperical experiments also show that the model is expressive enough to perform better while the bound being smaller. __We did not give an invalid bound like $accuracy \geq 0$, the model is approaching the bound.__
>
> >__W5__: In Corollary 3.1, the related equation is also a form of infimum. So, even if the right-hand side of equation 4 is equal to 0, that doesn't mean the left-hand side of equation 4 is equal to 0. Because the right side of equation 4 is the lower bound of the left.
>
> __Answer__: Indeed, a lower right-hand side of equation 4 does not enforce a better performance, but it shows that the model could potentially perform better. And we further design an algorithm to lower the right-hand of equation 4, experiments show that it does help, so the lower bound is not meaningless, and it could guide algorithm design.

---

> ### Author Response · Authors · 2023-11-19
> **Response to Reviewer vy1b Part 3**
>
> >__W6__: The improvement in performance is negligible.
>
> __Answer__: As we claimed in the paper, the two methods are mainly used to verify our theory, so we design two simple methods. But it does not mean it is not effective, the marginal improvement is mainly caused by causal hyperparameter selection, for example in the information augmentation, we fix the 4 dropout rate, learning rate, weight decay and all other hyperparameters shared with GRACE, and we just set the number to be masked as twice of the number to be unmasked. If we just fine the drop rate from 0.05 to 0.75 (we use the same drop rate for feature/edge drop of two views, but GRACE uses 4 different drop rate), we can get the result below:
>
> |             | Cora                 | CiteSeer             | PubMed               | DBLP                 | Amazon-P             | Amazon_C             |
> | :---:       |    :----:            |       :---:          |       :---:          |       :---:          |       :---:          |       :---:          |
> | GRACE       | 82.52$\pm$0.75       | 68.44$\pm$0.91       | 84.97$\pm$0.7        | 84.01$\pm$0.34       | 91.17$\pm$0.15       | 86.36$\pm$0.13       |
> | GRACE+I     | **83.78$\pm$1.08**   | **72.89$\pm$0.87**   | 84.97$\pm$0.5        | 84.80$\pm$0.47       | 91.64$\pm$0.21       | **87.67$\pm$0.07**   |
> | GRACE+S     | 83.61$\pm$0.85       | 72.83$\pm$0.47       | **85.45$\pm$0.5**    | **84.83$\pm$0.38**   | **91.99$\pm$0.17**   | 87.57$\pm$0.12       |
> | GCA         | 83.74$\pm$0.78       | 71.09$\pm$0.79       | 85.38$\pm$0.35       | 83.99$\pm$0.46       | 91.67$\pm$0.33       | 86.77$\pm$0.05       |
>
> So our methods achieve great improvement in performance, but to make the mutual information and $\delta_{aug}$ in Figure 2 comparable and show that our methods helps enlarge $\delta_{aug}$ while keep enough information, we use the same drop rate.
>
> >__W7__: My personal understanding of the two methods proposed in this paper is as follows: (1) From the perspective of information theory, the traditional GCL is based on learning, so that the mutual information between positive samples is as large as possible. This paper proposes that the mutual information between positive samples can be increased directly by data augmentation. (2) From the perspective of spectrum, the traditional GCL smoothen the spectrum through regularization, while this paper proposes that the mutual information between positive samples can be increased directly through data augmentation. Therefore, the motivation presented in this paper is not convincing.
>
> __Answer__: **Our paper does not mean mutual information between positive samples can be increased directly by data augmentation**, actually we propose to conduct stronger augmentation and the mutual information is decreasing because of stronger augmentation. Both of the two methods are trying to achieve a trade-off between the mutual information and strength of augmentation, the information augmentation directly protect the important features/edges from being masked and mask more those unimportant ones to prevent $\delta_{aug}$ from dropping sharply. The spectrum augmentation is also intended to improve $\delta_{aug}$ by modify the spectrum of graph. **So our two methods aims to achieve a trade-off between strength of augmentation ($\delta_{aug}$) and the mutual information**, one tries to increase mutual information while preventing $\delta_{aug}$ from dropping, and the other tries to increase $\delta_{aug}$ while prevent the mutual information from dropping.

---

> > ### Comment · Reviewer_vy1b · 2023-11-22
> > **Response to authors**
> >
> > After reading the rebuttal, I still have my doubts about the low bound. Also, the performance gain is weak. Thus, I keep my initial rating.

---

> > > ### Author Response · Authors · 2023-11-23
> > > **Response to Reviewer vy1b**
> > >
> > > Thanks for your suggestions, in our opinion, the lower bound represents the potential of the model. With a smaller lower bound, the model could potentially achieve better performance, for example, if there exists two models with $loss \geq 0.7$ and $loss \geq 0.3$ respectively, the latter one would be prefered as it is more likely to perform better.
> > >
> > > Also the weak performance gain is mainly caused by fixed hyperparameters. To show that our methods could maintain mutual information while increasing positive pair difference $\delta_{aug}$ (Figure 2), we use the same drop rate. But if we finetune the drop rate, the performance would be much better as shown below:
> > > |             | Cora                 | CiteSeer             | PubMed               | DBLP                 | Amazon-P             | Amazon_C             |
> > > | :---:       |    :----:            |       :---:          |       :---:          |       :---:          |       :---:          |       :---:          |
> > > | GRACE       | 82.52$\pm$0.75       | 68.44$\pm$0.91       | 84.97$\pm$0.7        | 84.01$\pm$0.34       | 91.17$\pm$0.15       | 86.36$\pm$0.13       |
> > > | GRACE+I     | **83.78$\pm$1.08**   | **72.89$\pm$0.87**   | 84.97$\pm$0.5        | 84.80$\pm$0.47       | 91.64$\pm$0.21       | **87.67$\pm$0.07**   |
> > > | GRACE+S     | 83.61$\pm$0.85       | 72.83$\pm$0.47       | **85.45$\pm$0.5**    | **84.83$\pm$0.38**   | **91.99$\pm$0.17**   | 87.57$\pm$0.12       |
> > > | GCA         | 83.74$\pm$0.78       | 71.09$\pm$0.79       | 85.38$\pm$0.35       | 83.99$\pm$0.46       | 91.67$\pm$0.33       | 86.77$\pm$0.05       |
> > >
> > > GRACE+I and GRACE+S reachs comparable even better performance with GCA, but GCA finetunes 4 different drop rates (the drop on edge and feature on both views) while we use only one drop rate. So our model could achieve great performance gain.

---

### Official Review · Reviewer_yxaQ · 2023-11-02

**Soundness:** 3 good
**Presentation:** 3 good
**Contribution:** 3 good
**Rating:** 6
**Confidence:** 3

**Summary:**

In this paper, the author proposes the relationship between data augmentation and downstream performance for graph contrastive learning. Inspired by this relationship, the author proposes a simple method to find an appropriate amount of data augmentation on a dataset.

In section 2 of the paper, the authors propose that a strong data augmentation can help lower the negative center similarity, which will increase the intra-class divergence. However, it is hard to say that the positive center similarity will increase as expected. Therefore, the lower negative similarity helps more in contrastive learning.  It is crucial to find better augmentation to help the model better generalize on downstream tasks rather than find a perfect alignment.

In section 3 of the paper, the authors propose methods to find the amount of data augmentation from two inspectives. From an information perspective, inspired by the previous methods, the gradient can used to determine the importance of the node. The important nodes should be changed less than other nodes. And unimportance nodes are free to mask. From a graph spectrum perspective, the linear combination of eigenvalues of adjacency matrix q can be considered as a signal. If q decreases, the change in spectrum is too much.

**Strengths:**

1: The author provides a new perspective to understand the data-augmentation and graph contrastive learning, which is worth since understanding how data augmentation works on graph contrastive learning will be important.

2: The author provides two methods to find proper augmentation amount from different perspectives.

3: These two methods are not complex and can easily fit to different graph contrastive learning methods.

**Weaknesses:**

1: In the figure 8, we can see that the information augmentation achieves highest positive center similarity. However, it also achieve highest negative similarity, which is not promising. Will that affect the final downstream performance?

2: From table 1, we can see that for same base methods, information augmentation performs better on certain datasets and spectrum augmentation performs better on others. If possible, can author discuss about how the data affect the efficiency of the data augmentation?

**Questions:**

Please refer to Weaknesses.

---

> ### Author Response · Authors · 2023-11-19
> **Response to Reviewer yxaQ**
>
> Thank you for taking the time to read our work and insightful comments and suggestions! We are happy to know that you think understanding how data augmentation works is important. Please find below our comments on the raised issues and questions:
>
> >__W1__: In the figure 8, we can see that the information augmentation achieves highest positive center similarity. However, it also achieve highest negative similarity, which is not promising. Will that affect the final downstream performance?
>
> __Answer__: Actually we find that positive center similarity and negative center similarity changes in similar pattern as shown in Figure 1, so if we want higher positive center similarity the negative center similarity also increases. In result, the infomation augmentation the highest positive and negative center similarity, also you can observe from Figure 8 that the spectrum augmentation also gets both lower positive and negative center similarity. But still they achieve better performance.
>
> __The reason why positive and negative center similarity tends to change in similar pattern might be augmentation overlap.__ Contrastive learning helps intra-class gathering mainly by augmentation overlap, for example, if two different nodes $v_i^0$, $v_j^0$ get the same augmentation $v^+$, then the best solution to contrative learning is $f(v_i^0)=f(v^+)=f(v_j^0)$. But if there is no augmentation overlap, __intra-class nodes are being separated just like inter-class nodes.__ And Saunshi et al. 2022 [1] propose that augmentation overlap is actually rare in contrastive learning. So actually intra-class nodes are treated the same with inter-class nodes, then posotive and negative center similarity changes in similar pattern.
>
> >__W2__: From table 1, we can see that for same base methods, information augmentation performs better on certain datasets and spectrum augmentation performs better on others. If possible, can author discuss about how the data affect the efficiency of the data augmentation?
>
> __Answer__: We believe that the for larger datasets, the information could still be well kept even after strong augmentation, so the information augmentation might not be that helpful as in small datasets. And the spectrum augmentation directly modify the spectrum which influence the whole graph, so it might help more significantly in larger graphs. The exception is AD GCL, as it aims to minimize the mutual information by neural network, so it loses lots of useful information and the information augmentation helps more significantly.
>
> [1]Saunshi N, Ash J, Goel S, et al. Understanding contrastive learning requires incorporating inductive biases[C]//International Conference on Machine Learning. PMLR, 2022: 19250-19286.

---

### Official Review · Reviewer_J3pe · 2023-11-03

**Soundness:** 3 good
**Presentation:** 2 fair
**Contribution:** 3 good
**Rating:** 6
**Confidence:** 3

**Summary:**

The paper investigates the impact of alignment usually implemented in Graph Contrasive Learning. Specifically, they find Graph Contrastive Learning (GCL) approach significantly contributes to downstream tasks primarily by effectively distinguishing between different classes, rather than aggregating nodes of the same class. The conventional belief that perfect alignment and augmented overlap, which treat all intra-class examples equally is challenged in this paper. To gain a deeper understanding of the role of augmentation in the contrastive learning process, the paper conducted extensive analysis of its impact on generalization and revealed that while perfect alignment, treating positive pairs equally, might enhance contrastive loss, it is detrimental to the overall generalization of the model.

**Strengths:**

S1: The study focuses on the influence of alignment in Graph Contrastive Learning. It emphasizes that the Graph Contrastive Learning (GCL) method significantly contributes to downstream tasks by effectively discerning between various classes, rather than consolidating nodes of the same class.

S2: This challenges the traditional assumption that perfect alignment and augmented overlap is beneficial in contrastive learning. In an effort to delve deeper into the role of augmentation in the contrastive learning process, the study extensively analyzes its impact on generalization. The paper then proposes a new method that smooths the spectrum to achieve the goal of optimization.

S3: The results reveal that while perfect alignment, treating positive pairs equally, may enhance contrastive loss, it ultimately hinders the overall generalization of the model.

**Weaknesses:**

W1: Theorem 2.5 explains why as the augmentation becomes stronger, negative center similarity decreases, (The answer is simultaneously we also want the inner class samples to be closer, and there is a tension between these two objectives. But this is almost always expected regardless of theorem 2.5). This does not mean that pulling closer the augmentations is completely harmful, as there is potentially a sweet point between pulling closer inner class samples and pushing away inter class samples that is optimal to the optimization, which is also (intuitively) dependent on the dimension of the features. I guess maybe the conclusion can be made more interesting and more surprising if it addresses where that sweet point is (if perfect alignment is not the sweet point).  It is even better if this sweet point is connected to the value of the threshold $\epsilon$. Currently, $\epsilon $ seems to be a heuristic and is not associated with the positive negative balance problem. Please kindly correct me if I have misunderstood your motivation.

W2: Empirical evidence shows very marginal improvements over the existing baselines, and there is no error bar reported. This concerns me in if the performance is stable and reliable. Please could you report error bars of the algorithms where each algorithm is implemented for several times and the result is averaged over the runs.

**Questions:**

Please see above for the questions.

---

> ### Author Response · Authors · 2023-11-19
> **Response to Reviewer J3pe**
>
> Thank you for taking the time to read our work and insightful comments and suggestions! We are happy to know that you appreciate the work. Please find below our comments on the raised issues and questions:
>
> >__W1__: There is potentially a sweet point between pulling closer inner class samples and pushing away inter class samples that is optimal to the optimization, which is also (intuitively) dependent on the dimension of the features. I guess maybe the conclusion can be made more interesting and more surprising if it addresses where that sweet point is (if perfect alignment is not the sweet point). It is even better if this sweet point is connected to the value of the threshold $\epsilon$. Currently, $\epsilon$ seems to be a heuristic and is not associated with the positive negative balance problem. Please kindly correct me if I have misunderstood your motivation.
>
> __Answer__: Thanks for your suggestion, It is a quite exciting idea. But $\delta_{aug}$ is defined using $\mathbb{E}||f(v^+)-f(v)||^2$, and we assume that when augmentation gets stronger $\delta_{aug}$ increases which means the alignment performance decreases. So we think that perfect alignment may not help downstream performance because only weak augmentation could be adopted (Appendix C.9 directly shows that stronger augmentation leads to worse alignment). So the sweet point correlates with the model, for example if the model is expressive enough to project all augmented samples the same no matter how strong the augmentation is, then the sweet point is exactly perfect alignment as $\mathbb{E}||f(v+)-f(v)||_2^2 = 0$, and this is the implicitly assumption of previous works.
>
> In regular scenes that the model is not so imaginary powerful, alignment performance still decreases with stronger augmentation. The right hand side of Theorem 2.5 is a quadratic function about $\delta_{aug}$, but $\mathrm{Var}(f(v^+)|y) \leq (\delta_{aug}+\sqrt{\mathrm{Var}(f(v_y^0)|y)})^2$, so the best $\delta_{aug}$ is related to  $\mathrm{Var}(f(v_y^0)|y)$ and it requires the class label information which is unknow while pretraining, so the sweet point can not be directly calculated according to our theory.
>
> But the theory could still help model analysing and design. You can gradually change the structure or hyperparameter of a model and observe how is the InfoNCE loss and positive pair difference $\delta_{aug}$ changing to understand how is the model working and why it helps. And if your model performs well on contrastive loss but fails to generalize to downstream tasks, you may need to conduct stronger augmentation while adopting other methods to keep InfoNCE loss steady.
>
> >__W2__: Empirical evidence shows very marginal improvements over the existing baselines, and there is no error bar reported. This concerns me in if the performance is stable and reliable. Please could you report error bars of the algorithms where each algorithm is implemented for several times and the result is averaged over the runs.
>
> __Answer__: Our results are actually the average of 10 runs, and it is stable and reliable, we have reported the error bar in Appendix C.10. Also the marginal improvements mainly blame to the hyperparameter, we follow all the hyperparameters of GRACE, but actually the two methods could bear different augmentation with GRACE as the two methods behaves differently when perform augmentation. But to show that our methods could indeed increase $\delta_{aug}$ while preserving information and make the result comparable with GRACE in Figure 2, we use the same drop rate with GRACE. And if we finetune the drop rate from 0.05 to 0.75 (we use the same drop rate for feature/edge drop of two views, but GRACE uses 4 different drop rate), we can get the result below:
> |             | Cora                 | CiteSeer             | PubMed               | DBLP                 | Amazon-P             | Amazon_C             |
> | :---:       |    :----:            |       :---:          |       :---:          |       :---:          |       :---:          |       :---:          |
> | GRACE       | 82.52$\pm$0.75       | 68.44$\pm$0.91       | 84.97$\pm$0.7        | 84.01$\pm$0.34       | 91.17$\pm$0.15       | 86.36$\pm$0.13       |
> | GRACE+I     | **83.78$\pm$1.08**   | **72.89$\pm$0.87**   | 84.97$\pm$0.5        | 84.80$\pm$0.47       | 91.64$\pm$0.21       | **87.67$\pm$0.07**   |
> | GRACE+S     | 83.61$\pm$0.85       | 72.83$\pm$0.47       | **85.45$\pm$0.5**    | **84.83$\pm$0.38**   | **91.99$\pm$0.17**   | 87.57$\pm$0.12       |
> | GCA         | 83.74$\pm$0.78       | 71.09$\pm$0.79       | 85.38$\pm$0.35       | 83.99$\pm$0.46       | 91.67$\pm$0.33       | 86.77$\pm$0.05       |
>
> We can observe that our methods achieve comparable even better performance with GCA, which is also a variant of GRACE.

---

> > ### Comment · Reviewer_J3pe · 2023-11-22
> > **Thanks for your response**
> >
> > Thanks for showing more empirical evidence. I will maintain my score.

---

> > > ### Author Response · Authors · 2023-11-23
> > > **Response to Reviewer J3pe**
> > >
> > > Thank you for appreciating our work and insightful suggestions!

---

### Official Review · Reviewer_yszY · 2023-11-04

**Soundness:** 3 good
**Presentation:** 3 good
**Contribution:** 3 good
**Rating:** 8
**Confidence:** 4

**Summary:**

This paper focuses on graph contrastive learning methodology and more specifically studies the augmentation methodology and how it affects the overall downstream performance of the model. The findings show that perfect augmentation overlap and alignment are not the key factor for contrastive learning. When the augmentation is better, the model performs better mainly because of inter-class separating rather than intra-class gathering brought by augmentation overlap. The authors analyze the contrastive process through information theory and graph spectrum theory and they propose two methods based on these theories to establish the relationship between downstream performance, contrastive learning loss and data augmentation. They use the proposed methodology to test state of the art contrastive learning methods and dataset, and the demonstrate the performance improvement.

**Strengths:**

1) The paper is on an interesting topic, and through extensive experiments, the authors show that the proposed theories can improve the performance of existing contrastive learning models in most cases.
2) The paper is well-written and easy to follow. The structure of the paper is very good, the authors describe the related works and the existing challenges and limitations. Then they describe their theories and then the provide with the theoretical background. At the end, they provide well designed experiments and they show the improvement on the performance on state of the art models in various datasets and downstream tasks.
3) The methodology is written to the detail. The experiments are sound and convincing.

**Weaknesses:**

1) The novelty of the proposed methodology is limited.
2) Some details on the experiments are missing. For example, what are the sizes of the datasets/train sets/validation sets/test sets? Are the reported performances tested for statistical significance? These are important details that will help the reader to be more convinced and make a stronger point on the improvement of the performances.
3) Another important detail on the experiments would be to add the run times, so that the reader would be able to see by adding the different augmentation strategies what is the extra computation time that it takes for those improvements?

**Questions:**

The authors can respond on my 2) and 3) comments on the weaknesses section above.

**Details Of Ethics Concerns:**

No concerns

---

> ### Author Response · Authors · 2023-11-19
> **Response to Reviewer  yszY**
>
> Thank you for taking the time to read our work and insightful comments and suggestions! We are happy that the peper is easy to follow to you and the topic is interesting. Please find below our comments on the raised issues and questions:
>
> >__W1__: The novelty of the proposed methodology is limited.
>
> __Answer__: The two proposed methods are simple, but the information augmentation directly mask more nodes to increase positive pair difference, which is always achieved implicitly by previous works, so the information augmentation reveals the reason why previous works help. And as far as we know, there is no previous work reveal that smoothen the graph spectrum could help contrastive learning, also we use the change of network parameter to judge is the model correctly trained and perform spectrum augmentation based on the status of the model, which is also a novel and effective method.
>
> >__W2__: Some details on the experiments are missing. For example, what are the sizes of the datasets/train sets/validation sets/test sets? Are the reported performances tested for statistical significance? These are important details that will help the reader to be more convinced and make a stronger point on the improvement of the performances.
>
>
> __Answer__: Thanks for your suggestion, we use 10% of nodes as train set and others as test setm and similar to GRACE, logistic regression is adopted as classifier. We have added more experiment details in Appendix C.1, and we report the statistical significance in Table 1 it shows that our methods actually improve the downstream performance, also we show the error bar of proposed methods on various datasets in Appendix C.10.
>
> >__W3__: Another important detail on the experiments would be to add the run times, so that the reader would be able to see by adding the different augmentation strategies what is the extra computation time that it takes for those improvements?
>
> __Answer__: Thanks for your suggestion, we have conducted experiments on the run times (seconds) as shown below and added it in Appendix C.10
> |       | Cora      | CiteSeer  | PubMed    | DBLP    | Amazon-P  | Amazon_C   |
> | :---: |:----:     | :---:     | :---:     |:---:    | :---:     |  :---:     |
> | GRACE | 8.02      | 10.08     | 62.37     |56.89    | 19.05     | 28.71      |
> |GRACE+I| 10.74     | 13.49     | 68.97     |62.82    | 22.67     | 29.61      |
> |GRACE+S| 9.61      | 12.46     | 78.11     |69.44    | 21.13     | 36.94      |
>
> we can observe that the information augmentation method achieve better performance with only few more time consuming, this is mainly because we do not calculate the importance of features/edges every epoch we only calculate it once and use the same importance for the following training. The spectrum augmentation consumes more time when dealing with large graphs because we explicitly change the spectrum and calculate the new adjacency matrix, which could be replaced by some approximation methods. But to prevent interference from random noise and show smoothing the spectrum works, we use no approximation.

---

### Official Review · Reviewer_KFkN · 2023-11-05

**Soundness:** 3 good
**Presentation:** 2 fair
**Contribution:** 2 fair
**Rating:** 6
**Confidence:** 4

**Summary:**

This paper explores the relationships between 1) augmentations and the difficulty of training contrastive loss, and 2) augmentations and the generation ability. It highlights that good alignment helps contrastive learning but hurts generation. The paper also proposes GCL augmentations based on the perspectives of information theory and graph spectrum, and validates their effectiveness through experiments.

**Strengths:**

The relationships between augmentations, contrastive training and generation is a valuable topic; The experiments are sufficient; The analysis is informative (though sometimes confusing in writing).

**Weaknesses:**

**W1.** Lack of related work. I note that you positioned the related work in the appendix. However, given that you mentioned other works (like Wang et al. 2022b, Saunshi et al. 2022) many times in the paper, an explicit and focused discussion on the differences between your work and others is expected.

**W2.** Some definitions are unclear, resulting in harder understanding. For example,

1) *augmentation overlap.* You give an intuitive explanation as “support overlap between different intra-class samples”.  However, it is a specific term based on the analysis tool *augmentation graph*, first proposed by Wang et al 2022b. I think a formal definition is needed. The same is true for *perfect alignment*.

2) $\delta_{aug}$. I notice that it first occurs in Assumption 2.1 without any explanation. Considering it plays a key role in the following analysis, I think at least an intuitive explanation is needed here. Likelywise, the $\omega_y$ in $L_{CE}$ and $M$ in $L_{NCE}$.

**W3.** Lack of novelty for proposed augmentations. For the information-theory augmentation, it is quite similar with that proposed in Wei et al. 2023 (GCS). Similarly, the graph-spectrum augmentation is somewhat based on Liu et al. 2022 (GAME). However, both GCS and GAME are not chosen as a baseline in the experiments. As claimed, “this algorithm is primarily intended for theoretical verification”, I think the contribution is more like explaining existing augmentations by theory.

**W4.** I have a concern that label-consistency is more hard to guarantee on graphs (unlike images that can be judged manually). Therefore. it is difficult to figure out what kind of graph augmentations is strong enough while not perturbs the labels (i.e. satisfying the basis Assumption 2.1). Considering that, maybe experiments on images are more convincing.

**Questions:**

**Q1.** In Para3 of Introduction, you said “But it works the same in other fields.” Given that $L_{NCE}$ and theorems in the paper are all based on graph nodes, I wonder if they can generate to other domains like images. It is worth mentioning that graph contrastive learning and vision contrastive learning can not be confused with each other, see [1].

**Q2.** Questions about Figure 1.

1) According to Theorem 2.3 and your statement “Therefore, from Inequality (2), …, to be lower.”, it seems like the tendencies of positive and negative center similarity are different, but i did not see such phenomenon in Figure 1.

2) The tendencies are not consistent across 4 datasets. For examples, accuracy tendencies on only 2/4 datasets (Cora, DBLP) are similar, i.e. increasing first and then decreasing. Considering the diversity of graph benchmarks, can you show more evidence on other benchmarks besides citation networks?

3) More experimental details are needed, including the backbone, data splitting. Especially, no current GCL methods under widely-adopted settings can reach nearly 100% accuracy on these citation benchmarks, as far as I know.

**Q3.** According to Corollary 3.1, you said “the best augmentation would be one that maximize the mutual information and positive pair difference”. It seems to be contrary to the InfoMin principle [2], which claims that good views should minimize $I(v_1,v_2)$ while keep the label unchanged (i.e. $I(v_1,y)=I(v_2,y)=I(x,y)$). Can you give some explanations for the difference?

**Q4.** Minors.  In section 2.3, “Theorem 2.5 explains why as the augmentation becomes stronger, negative center similarity decreases”. Maybe it should be Theorem 2.3.

[1]. Guo et al. Architecture Matters: Uncovering Implicit Mechanisms in Graph Contrastive Learning. NeurIPS 2023.

[2]. Tian el al. What Makes for Good Views for Contrastive Learning? NeurIPS 2020.

---

> ### Author Response · Authors · 2023-11-19
> **Response to Reviewer KFkN Part 1**
>
> Thank you for taking the time to read our work and insightful comments and suggestions! We are glad to know that the relationship between augmentation, contrastive learning and generation is interesting to you. Please find below our comments on the raised issues and questions:
>
> >__W1__: Lack of related work. I note that you positioned the related work in the appendix. However, given that you mentioned other works (like Wang et al. 2022b, Saunshi et al. 2022) many times in the paper, an explicit and focused discussion on the differences between your work and others is expected.
>
> __Answer__: The main difference between our work and previous ones is that, previous works like Wang et al. 2022b regard perfect alignment and augmentation overlap as the key to success, but Saunshi et al. 2022 argues that augmentation overlap might be rare in contrastive learning. We propose that perfect alignment requires for weak augmentation and it tends to result in bad downstream performance while stronger augmentation leads to better downstream performance but inevitably hinders the alignment performance. So perfect alignment does not imply a good contrastive model.
>
> >__W2__: Some definitions are unclear, resulting in harder understanding.
>
> __Answer__: Thanks for your suggestion, we have added more explanation words in section 2.2 of our paper!
>
> >__W3__: Lack of novelty for proposed augmentations. For the information-theory augmentation, it is quite similar with that proposed in Wei et al. 2023 (GCS). Similarly, the graph-spectrum augmentation is somewhat based on Liu et al. 2022 (GAME). However, both GCS and GAME are not chosen as a baseline in the experiments. As claimed, “this algorithm is primarily intended for theoretical verification”, I think the contribution is more like explaining existing augmentations by theory.
>
> __Answer__: The two mathods are indeed trying to explain existing augmentations by theory. But they do not follow existing methods, we propose the information augmentation directly by Corollary 3.1, and the information based methods just use saliency to mearuse the importance of features/edges but the main architecture is totally different with GCS, GCS use the saliency to split the graph into the semantic and environment subgraph and maximize the mutual information between semantic and minimize the mutual information between the environment. But we only use the saliency to indicate the importance and then choose to mask the feature/edge or unmask it by the saliency.
>
> And the GAME rule requires to change high-frequency information while capturing the common low-frequency information by data augmentation. However, what we propose is to smooth the eigenvalues of adjacency matrix. For a graph with adjacency matrix $A$ and two augmentation views $A',A''$, and $\lambda,\lambda',\lambda''$ are the eigenvalue of $A,A',A''$, respectively. The GAME rule requires that for $\lambda_i \in [0,1]$ and $\lambda_j \in [-1,0]$, the augmentation should satisfy $|\lambda_i'-\lambda_i''| > |\lambda_j'-\lambda_j''|$ (Figure 7 of GAME). But we want that for any $\lambda_i$, the augmentation spectrum $\lambda_i' \cdot \lambda_i''$ should be smaller, __so different from GAME, we treat $\lambda_i \in [0,1]$ and $\lambda_j \in [-1,0]$ the same__, and then we use the parameter $\theta$ to gradually lower $\lambda' \cdot \lambda''$ and smooth the spectrum. Also the GAME is actually conclusion of emperical results that low-frequency information matters more in GNN, but for graph with heterophily, high-frequency information is the key. __So GAME is emperical conclusion on some dataset, and our method is derived by theory and could be adopted on various of datasets.__
>
> Also we have conducted experiments on GCS and GAME and the result show that our methods outperform them and the results are shown in Table 1.

---

> ### Author Response · Authors · 2023-11-19
> **Response to Reviewer KFkN Part 2**
>
> >__W4__: I have a concern that label-consistency is more hard to guarantee on graphs (unlike images that can be judged manually). Therefore. it is difficult to figure out what kind of graph augmentations is strong enough while not perturbs the labels (i.e. satisfying the basis Assumption 2.1). Considering that, maybe experiments on images are more convincing.
>
> __Answer__: Actually the label-consistency is easier to guarantee on graphs. Although the images can be judged manually, it is impossible to actually conducted manual judgement on large datasets, and augmentations like crop can easily change the class label, so the training could be confused. In graph contrastive learning, the augmentation may also change some nodes significantly, but the random augmentation is repeated every epoch, so even for some epoch, a node loses too much information, in other epochs, the node could still be correctly trained, so every node could be correctly trained unless the drop rate is too strong.
>
> To verify this, we first train a well-defined model (the GCL model with best performance) and use the prediction as the true label probability, then we change the drop rate from 0.05 to 0.95, predict the label probability and calculate the KL divergence with true label probability. We show the KL divergence when the drop rate is 45% below and other results could be seen in Appendix C.9
> |       | Cora      | CiteSeer  | PubMed    | DBLP    | Amazon-P  | Amazon-C   |
> | :---: |:----:     | :---:     | :---:     |:---:    | :---:     |  :---:     |
> | 45%   | 0.06      | 0.04      | 0.05      |0.01     | 0.03      | 0.01       |
>
> We can observe that the label is consistent with 45% drop rate, and from Figure 1, we can tell that the downstream performance tends to decrease when drop rate is greater than 0.5, and our theory considers the situation when downstream performance is increasing and GCL methods are not likely to use augmentation stronger than 45% drop, so the label consisteny assumption is easy to hold.
>
> Also we conduct experiments on images, and the result shows that it also aligns with our expectation that downstream performance correlates with inter-class separating stronger. Please refer to answer to Q1 for further discussion.
>
>
> >__Q1__: In Para3 of Introduction, you said “But it works the same in other fields.” Given that $L_{NCE}$ and theorems in the paper are all based on graph nodes, I wonder if they can generate to other domains like images. It is worth mentioning that graph contrastive learning and vision contrastive learning can not be confused with each other.
>
> __Answer__: Actually the theoretical is not closely related to graph, we only conduct experiments on graph datasets because the magnitude of augmentation on graphs (drop rate of feature/edge) is easy to control and. And we have conducted experiments on images to extend our theory to images. We use the color distortion as data augmentation and change the color distortion strength and observe how the positive/negative center similarity changes as shown below, more results could be seen in Appendix C.8:
> | stl10 | 0.2       | 0.4       | 0.6       | 0.8     | 1.0       |
> | :---: |:----:     | :---:     | :---:     |:---:    | :---:     |
> | PCS   | 0.48      | 0.47      | 0.43      |0.47     | 0.47      |
> | NCS   | 0.11      | 0.09      | 0.05      |0.04     | 0.04      |
> | Acc   | 0.68      | 0.69      | 0.71      |0.73     | 0.73      |
>
> We can see that contrastive learning in CV also show similar tendency that negative center similarity (NCS) contributes to downstream performance, but different from GCL (graph contrastive learning), VCL (vision contrastive learning) also benefits from increasing positive center similarity (PCS) especially when the color distortion strength (CDS) is stronger. But still, decreasing NCS plays a more important role when CDS is not that strong, so our theory could also be extended to VCL.

---

> ### Author Response · Authors · 2023-11-19
> **Response to Reviewer KFkN Part 3**
>
> >__Q2__:
> >
> >1. According to Theorem 2.3 and your statement “Therefore, from Inequality (2), …, to be lower.”, it seems like the tendencies of positive and negative center similarity are different, but i did not see such phenomenon in Figure 1.
> >
> >2. The tendencies are not consistent across 4 datasets. For examples, accuracy tendencies on only 2/4 datasets (Cora, DBLP) are similar, i.e. increasing first and then decreasing. Considering the diversity of graph benchmarks, can you show more evidence on other benchmarks besides citation networks?
> >
> >3. More experimental details are needed, including the backbone, data splitting. Especially, no current GCL methods under widely-adopted settings can reach nearly 100% accuracy on these citation benchmarks, as far as I know.
>
> __Answer__:
> 1. Actually we did not mean that the tendencies of positive and negative center similarity are different, we want to tell that the positive center similarity is hard to predict as the $\delta_{aug}$ increases and $\delta_{y^+}$ decreases, but the negative similarity are likely to decrease as $\delta_{aug}$ and $\delta_{y^-}$ both increases. __So we imply that as the augmentation gets stronger and downstream performance getting better, negative center similarity should be decreasing, and the positive center similarity may not be increasing.__ In Figure 1, the __PCS and NCS are mostly decreasing together when at the begining which is consistent with our expectation.__ The PCS and NCS show similar tendency may blame to rare augmentation overlap proposed by Saunshi et al. 2022, without augmentation overlap, the intra-class nodes are treated the same with inter-class nodes, so all nodes are being separated further then PCS and NCS both decreases.
>
> 2. Of course, we conduct further experiments on the shopping network (Amazon-Photo and Amazon-Computers), graph with heterophily (Chameleon) and the Coauthor network (Coathor-CS), they all show similar tendency and the results are shown in Appendix C.8. Actually the dataset CiteSeer and PubMed also follows the schema as we set the start point of dropout of 0.05 rather than 0, because 0 means no augmentation and contrastive could learn nothing about the graph. So we also conduct experiments when the drop rate is small in Figure 12. Experimental results show that with the augmentation being stronger the downstream performance is likely to increase at the begining and decreases later. And they both show that the downstream performacne correlates with negative center similarity stronger.
>
> 3. We use a 2-layer GCN as the backbone and use 10% of data for training and the rest for testing, we provide further experimental detail in Appendix C.1. And we also did not reach nearly 100% accuracy. As we aim to to show the tendency of positive/negative center similarity, __Figure 1 shows the Min-Max Normalized data, so the maximization is 1.__
>
>
> >__Q3__: According to Corollary 3.1, you said “the best augmentation would be one that maximize the mutual information and positive pair difference”. It seems to be contrary to the InfoMin principle, which claims that good views should minimize $I(v_1,v_2)$ while keep the label unchanged (i.e. $I(v_1,y)=I(v_2,y)=I(x,y))$. Can you give some explanations for the difference?
>
> __Answer__: The InfoMin principle actually wants to minimize $I(v_1,v_2)$ while preserve as much downstream information as possible. However, downstream tasks is unknown while pretraining, so this is acutally impossible to achieve. Our theory indicates that the augmentation should be strong while preserve as much information as possible, so we do not want to maximize the mutual infomation as stronger augmentation would inevitably leads to smaller mutual information. We want to achieve a trade-off between augmentation and mutual information, and we think the best trade off would be the one satisfying InfoMin which means the augmentation gets rid of all useless information and keeps the downstream related ones. __So InfoMin propose the ideal augmentation which can not be achieved, and we propose a actual target to train a better model.__
>
> >__Q4__: Minors. In section 2.3, “Theorem 2.5 explains why as the augmentation becomes stronger, negative center similarity decreases”. Maybe it should be Theorem 2.3.
>
> __Answer__: Thanks for your reminder, we have corrected this in the revised vision.

---

> ### Comment · Reviewer_KFkN · 2023-11-22
>
> I thank the authors for their responses and clarifications, and for addressing some critical comments by doing additional experiments. I have remaining concerns below:
>
> 1. I admit that the two proposed augmentations are different from GCS and GAME. However, they indeed lack novelty, otherwise, there is no need to distinguish the detailed differences as you did. Besides the empirical comparison added in Table 1,  presenting theoretical advantages (e.g., via Corollary 3.3) over other methods may strengthen your arguments.
> 2. About the label consistency on graphs, you show the label is consistent with a 45% drop rate under some datasets. I'm afraid that it is due to the robustness of (relatively) large social networks. However, for graphs like molecular or protein, changes in connections are prone to disturb its properties, as shown in Table 2 of [1].
> 3. If you insist that “The theoretical is not closely related to graph”, I think more theoretical explanations should be given. For instance, assume the i.i.d distribution of $v_i$, so $v_i$ can also be viewed as a randomly sampled image instance. (just a not rigorous example :) ). And I did not find more results on images in Appendix C.8. Please check for possible omissions.
> 4. I recommend revising the part about Figure 1: 1) The current explanation on the tendency is confusing, and directly causes my questions in the first round rebuttal; 2) The diversity of datasets (as you have added above); 3) Clarify the illustration of Figure 1, especially the ”normalized values”.
>
> [1]. Puja et al. Augmentations in Graph Contrastive Learning: Current Methodological Flaws & Towards Better Practices. WWW 2022.
>
> Score update: 5 → 6.

---

> > ### Author Response · Authors · 2023-11-23
> > **Response to Reviewer KFkN**
> >
> > Thank you for appreciating our work and increasing the score, and we are sincerely greatful for your suggestions! Please find below our comments on the raised issues and questions:
> >
> > >__Concern 1__: Theoretical advantages of proposed method.
> >
> > __Answer__: Thanks for your suggestion, we conducted experiments on GCS and GAME then report their mutual information and positive pair difference $\delta_{aug}$ below to show the superiority of our methods below:
> >
> > |   MI  | Cora      | CiteSeer  | PubMed    | DBLP    | Amazon-P  | Amazon-C   |
> > | :---: |:----:     | :---:     | :---:     |:---:    | :---:     |  :---:     |
> > | GRACE | 1.95      | 1.05      | 1.25      |1.32     | 2.46      | 2.52       |
> > |GRACE+I| 1.93      | 1.05      | 1.21      |1.30     | 2.30      | 2.47       |
> > |GRACE+S| 1.84      | 1.04      | 1.23      |1.29     | 2.33      | 2.46       |
> > |GAME   | 1.75      | 1.00      | OOM       |1.26     | 1.98      | 2.17       |
> > |GCS    | 1.97      | 1.04      | 1.33      |1.41     | 2.45      | 2.52       |
> >
> > |$\delta_{aug}$| Cora      | CiteSeer  | PubMed    | DBLP    | Amazon-P  | Amazon-C   |
> > | :---:        |:----:     | :---:     | :---:     |:---:    | :---:     |  :---:     |
> > | GRACE        | 0.34      | 0.20      | 0.23      |0.20     | 0.23      | 0.29       |
> > |GRACE+I       | 0.35      | 0.29      | 0.26      |0.22     | 0.24      | 0.32       |
> > |GRACE+S       | 0.39      | 0.23      | 0.26      |0.24     | 0.27      | 0.33       |
> > |GAME          | 0.43      | 0.28      | OOM       |0.24     | 0.27      | 0.37       |
> > |GCS           | 0.18      | 0.14      | 0.17      |1.41     | 2.45      | 2.52       |
> >
> > We can observe from the above tables that our methods achieve comparable mutual information while enlarge the positive pair difference $\delta_{aug}$, so the downstream performance is satisfying. However, the GAME conducts augmentation without considering the information loss, and loses too much information leading to worse performance. And GCS considers the maintain more task related information but did not notice the importance of stronger augmentation, so it also results suboptimal performance.
> >
> > >__Concern 2__: Augmentation on small graphs like molecular or protein may change the label as methoned by Puja et al. 2022
> >
> > __Answer__: It is true that larger networks tends to be more robust and can bear stronger augmentation, but we do not think this matters. The paper considers the situation when the label is consistent after augmentation since stronger augmentation would change the label information and perform terrible. __Smaller graphs may satisfy label consistency with at most 5% drop rate rather than 45% (for random mask), but what we want is the same: stronger augmentation and more information, and the most popular way to achieve it is context-aware augmentation__
> >
> > According to our theory, contrastive learning on molecular graphs is likely to performance badly because of weak augmentation (or label inconsistency). And according to Corollary 3.1 of our paper, we should maintain the information while increase the magnitude of augmentation. So when the augmentation is weak and mutual information is high, we need to conduct stronger augmentation, when augmentation is strong and mutual information is low, we need to capture more information (by weaker augmentation or context-aware augmentation). And what Puja et al. 2022 discussed is actually the latter situation that regular augmentation is too strong for molecular graphs, so they conduct context-aware graph augmentation to maintain more information.
> >
> > __Therefore, what Puja et al. 2022 did actually could be well explained by our theory.__
> >
> > >__Concern 3__: Relationship between theoretical results and graph data.
> >
> > __Answer__: Thanks for the suggestion, in our opinion, node embeddings in a graph could also be regared as i.i.d, and contrastive learning in vision and graph mainly differs in the augmentation methods and the encoder. Our analysis use $\delta_{aug}=\mathbb{E}||f(v)-f(v^+)||^2$ to represent the augmentation which is unrelevant to graph, and the encoders are both represented by $f$, so the theoretical results are not closely related to graph (except section 3.2 where we use graph spectrum to analyse the encoder and design algorithm). Please correct us if we have any misunderstandings.
> >
> > We add Table 5 in Appendix C.9 to illustrate how is the positive/negative center similarity and downstream performance changing with different color distorition strength on cifar10 and stl10.
> >
> > >__Concern 4__: Revision on the part about Figure 1
> >
> > __Answer__: Thanks for your suggestion, we have updated the explanation in the revised paper.
> >
> > Thank you again for you insightful suggestions!

---

### Author Response · Authors · 2023-11-20
**To All Reviewers: Summary of Our Revisions**

Thank you for taking the time to read our work and insightful comments and suggestions! We have updated our paper and the summary of revisions are as follows:
1. We add more experiments on the tendency of PCS/NCS and downstream performance including experiemnts on images to verify our findings in Appendix C.8.
2. We use statistical significance and the error bar to show that our methods actually ourperforms others.
3. We show the alignment performance is affected by stronger augmentation and the label consistency assumption is reasonable in graph in Appendix C.9.
4. More experimetal details like the run time and more explanation words are added.

---

### Meta-Review · Area_Chair_Bt1W · 2023-12-09

**Metareview:**

The paper offers a new perspective on GCL, focusing on how perfect alignment and augmentation impact model performance. However, it faces significant issues, including unclear theoretical explanations, lack of novelty in proposed methods, and insufficient experimental details. Additionally, the concerns about label consistency and the negligible improvement in performance raise questions about the practical significance of the findings. To strengthen the paper, the authors should address these concerns, clarify theoretical aspects, and provide more comprehensive and detailed experimental evidence.

**Justification For Why Not Higher Score:**

While the paper addresses an interesting aspect of graph contrastive learning, significant issues raised by the reviewers necessitate this decision.

Questions Regarding Novelty: The novelty of the proposed methods is questionable, as they appear similar to existing approaches. This similarity diminishes the paper's contribution to advancing the field.

Concerns About Label Consistency on Graphs: The feasibility of ensuring label consistency on graphs, crucial for the paper's methodology, is not convincingly addressed, raising doubts about the practical applicability of the approach.

Theoretical Interpretation Issues: There are significant concerns about the interpretation and application of theorems and equations, which could impact the theoretical soundness of the paper.

Negligible Improvement in Performance: The reported improvement in performance is minimal, questioning the practical significance and impact of the proposed methods.

**Justification For Why Not Lower Score:**

NA

---

### Decision · Program_Chairs · 2024-01-16

Reject